



# Atmospheric Blocking and Weather Extremes over the Euro-Atlantic Sector - A Review

Lisa-Ann Kautz[1], Olivia Martius[2], Stephan Pfahl[3], Joaquim G. Pinto[1], Alexandre M. Ramos[4], Pedro M. Sousa[4,5], and Tim Woollings[6]

[1]Karlsruhe Institute of Technology, Institute of Meteorology and Climate Research, Karlsruhe, Germany
[2]Institute of Geography, and Oeschger Centre for Climate Change Research, University of Bern, Bern, Switzerland
[3]Institute of Meteorology, Freie Universität Berlin, Berlin, Germany
[4]Instituto Dom Luiz (IDL), Faculdade de Ciências, Universidade de Lisboa, 1749-016 Lisboa, Portugal
[5]Instituto Português do Mar e da Atmosfera (IPMA), 1749-077 Lisboa, Portugal
[6]Atmospheric, Oceanic and Planetary Physics, University of Oxford, Oxford, UK

**Correspondence:** Lisa-Ann Kautz (lisa-ann.kautz@kit.edu)

**Abstract.** The physical understanding and timely prediction of extreme weather events are of enormous importance to society regarding associated impacts. In this article, we highlight several types of weather extremes occurring in Europe in connection with a particular atmospheric flow pattern, known as atmospheric blocking. This flow pattern effectively blocks the prevailing westerly large-scale atmospheric flow, resulting in changing flow anomalies in the vicinity of the blocking system and persistent

conditions in the immediate region of its occurrence. Blockings are long-lasting, quasi-stationary, self-sustaining systems that occur frequently over certain regions. Their presence and characteristics have an impact on the predictability of weather extremes and can thus be used as potential indicators. The phasing between the surface and the upper-level blocking anomalies is of major importance for the development of the extreme event. In summer, heat waves and droughts form below the blocking anticyclone primarily via large-scale subsidence that leads to cloud-free skies and thus, persistent longwave radiative warming

of the ground. In winter, cold waves that occur during atmospheric blocking are normally observed downstream or south of these systems. Here, horizontal advection of cold air masses from higher latitudes plays a decisive role. Extreme snowfall can also occur with the lower temperatures, indicating a shift of the storm track due to the blocking system. Such a shift is also crucial in the connection of blocking with wind and precipitation anomalies in general. Due to this multifaceted linkages, compound events are often observed in conjunction with blocking conditions.

## 1  Introduction

Weather extremes have a great significance for society, as they pose a threat to human life and can result in enormous economic damage and disruption. The heat wave in 2010, which affected Eastern Europe and large parts of Russia, is a prominent example of such an event (Barriopedro et al., 2011; Grumm, 2011). Heat records were broken in many areas and Moscow recorded temperatures of almost 40 °C. The heat wave was associated with an extreme drought resulting in thousands of forest

fires that damaged agriculture (Witte et al., 2011). The forest fires also caused air pollution associated with health risk. Another example for a high impact weather event is the cold spell at the beginning of 2012 that affected Europe (de'Donato et al.,





2013; Demirtaş, 2017). Temperatures around $-40\,°C$ were observed in Russia and parts of Scandinavia, but also in Southern European countries like Greece, temperatures fell below $-20\,°C$. In addition to the low temperatures, parts of Southeastern Europe also experienced heavy snowfall, which strongly affected the transport sector (Davolio et al., 2015). In total, 650 deaths are attributed to this cold spell (DWD, 2012). Besides cold and heat waves, Europe is affected by other types of high impact weather events, like floods. In autumn 2000, several heavy precipitation events led to flooding in Switzerland and parts of Southern Germany (Lenggenhager et al., 2019). In Switzerland, basements as well as streets were flooded and some roads had to be completely closed due to the danger of landslides. A train also derailed as a result of a landslide. As different as these examples for weather extreme events might have been, they have something in common. Namely the prevailing large-scale flow pattern in the troposphere (layer up to 10-12 km altitude), which was strongly influenced by atmospheric blocking (hereinafter referred to as *blocking*).

Blocking systems can be described as long-lasting, quasi-stationary and self-sustaining tropospheric flow features that are associated with a large meridional (north-south) flow component and thus, an interruption and/or deceleration of the zonal westerly flow in the midlatitudes (however, simultaneously, a strong zonal flow may be found north and south of the blocking systems) (Liu, 1994; Nakamura and Huang, 2018). Thus, their onset and decay phases are characterized by transitions form a more zonal to a more meridional flow pattern and vice versa, which is challenging for forecast models (Frederiksen et al., 2004). In addition, blockings are associated with complex dynamics that link different spatial and temporal scales and affect both their internal evolution and interactions with the flow environment (Shutts, 1983; Lupo and Smith, 1995). Blocking systems extend vertically across the whole troposphere and are at the surface typically associated with large high pressure systems (Schwierz et al., 2004), although the occurrence of heat lows below the blocking ridge is also observed. They occur both over the oceans and over land masses (Barriopedro et al., 2006; Tyrlis and Hoskins, 2008) and may cause extreme weather events, whereas the type of extreme event is sensitive to the exact location of the blocking system (Brunner et al., 2018). Moreover, different surface extremes at different locations (and sometimes at the same time) can be caused by the same blocking system (Lau and Kim, 2012).

Since the 1950s, the phenomenon of blocking has been studied by atmospheric scientists. Both the synoptic time scales (Colucci, 1985; Crum and Stevens, 1988) and the climate perspective (Nabizadeh et al., 2019) are of particular interest. Many studies are also available on the dynamical aspects (Berggren et al., 1949; Steinfeld and Pfahl, 2019), whereby interactions with different scales (Lupo and Smith, 1995; Luo et al., 2014) or with other flow features (Shutts, 1983; Shabbar et al., 2001) are considered. An important branch of research is concerned with how well blocking systems can be predicted (Bengtsson, 1981; Matsueda, 2009) or how blocking systems affect the quality of weather predictions (Quandt et al., 2017; Ferranti et al., 2018). A review on blocking, in particular with regard to the projected changes in blocking occurrence and characteristics under climate change, was provided by Woollings et al. (2018). A general review has recently been published by Lupo (2020). Although the range of studies dealing with blocking is wide, there is no summary yet that specifically addresses the influence of this phenomenon on surface weather extremes. This paper focuses on this gap. In doing so, we both highlight how specific





these influences on each type of extreme can be and make a connection between them.

The article is structured as follows. In Sec. 2, a brief summary of blocking characteristics addressing also important dynam-
ical features as well as the predictability of blocking is provided. Sec. 3 deals with temperature extremes, while hydrological
extremes are discussed in Sec. 4. For both types of extremes, an overview, a description of the relevant dynamics and several
case studies are presented in both sections. Sec. 5 provides an overview on other extremes related to extreme winds and ad-
dresses compound events. In Sec. 6, the issue of predictability is revisited but with a focus on the impact of blocking on the
predictions of surface extreme events. In Sec. 7, we address changes in blocking and weather extremes due to climate change.
An outlook and possible research perspectives are presented in the last section.

## 2 Atmospheric blocking

### 2.1 Definition and Characteristics

Blocking systems are characterized by their persistence, quasi-stationarity and self-preservation. Although these characteristics
are common to most blocking systems, there is no unique definition owing to the rich diversity in synoptic structure. Following
the pioneering study of Rex (1950), many consider an essential feature of blocking to be a sharp transition from a zonal to
meridional flow pattern, as the jet is typically split into two branches around the system. Blocking systems generally fall into
the following categories, examples of which are shown in Woollings et al. (2018):

**Rex or dipole blocks** consist of an anticyclone lying poleward of a cyclone. These are closely linked to the breaking of
atmospheric Rossby waves which acts to reverse the usual meridional flow gradients (Hoskins et al., 1985; Pelly and
Hoskins, 2003a). Wave breaking can take an anticyclonic or cyclonic form and both lead to similar meridional dipole
structures (Weijenborg et al., 2012; Masato et al., 2013a).

**Omega blocks** are characterized by a huge anticyclone flanked by an upstream and a downstream cyclone leading to an
omega-shaped flow pattern (Ser-Giacomi et al., 2015). Although most common in the Pacific/North America sector, they
do also occur over Eurasia.

**Amplified ridges** without any closed contours (in e.g. $500\,\mathrm{hPa}$ geopotential height) are also able to block the zonal flow and
to lead to a dominating meridional flow component, especially in summer (Sousa et al., 2018). They are more common
at lower latitudes.

A large array of blocking indices has been developed, each designed to capture one or more of the structures within this
diversity. One approach is to identify blocking as a long-lasting anomaly, for example, in the potential vorticity (PV) field at
$320\,\mathrm{K}$ (associated with a negative anomaly) (Schwierz et al., 2004). Another way to identify blocking is to detect the reversal





in horizontal gradients, for example, in the 500 hPa geopotential height (Tibaldi and Molteni, 1990; Scherrer et al., 2006) or the potential temperature at 2 PVU (Pelly and Hoskins, 2003a). Some indices are one-dimensional (e.g. the index of Pelly and Hoskins (2003a) is calculated along the so-called "central blocking latitude"), while others present blocking patterns as two-dimensional structure (Masato et al., 2013a). In some studies, additional spatial and temporal criteria addressing blocking duration (number of blocked days) and the extension of blocking systems are considered (Barnes et al., 2012). A recent work by Sousa et al. (2021) has explored a conceptual model for the life-cycle of blocks, considering the dynamical process of incipient subtropical ridges transitioning towards an Omega block, trough wave breaking, and towards the mature phase of a fully secluded Rex block. A detailed overview of blocking detection indices is provided by Barriopedro et al. (2010). Besides these more synoptic descriptions (Liu, 1994), blocking can also be described by local finite amplitude wave metrics (Chen et al., 2015; Huang and Nakamura, 2016; Martineau et al., 2017).

Blocking in the northern hemisphere occurs predominantly for specific regions (Barriopedro et al., 2006; Tyrlis and Hoskins, 2008), both over land and oceans (Fig. 1). Over land, blocking is preferably found over a region reaching from Europe (especially over Scandinavia) (Tyrlis and Hoskins, 2008) to Asia (especially over the Ural mountains) (Cheung et al., 2013). Europe is identified as a dominant region of blocking in most indices, due to the configuration of a strong, meridionally-tilted storm track upstream of a large landmass. Blocking also occurs frequently over Greenland with strong downstream impacts on Europe associated with the negative phase of the North Atlantic Oscillation (NAO) (Davini et al., 2012). Additionally, blocking occurs over Northern America, where it is also associated with extreme events, such as temperature or precipitation anomalies (e.g. the Gulf of Alaska blocking event in summer 2004 leading to abnormally high temperatures and less-than-normal precipitation) (Glisan, 2007; Whan et al., 2016). Many of these preferential areas for blocking occurrence tend to represent an extension of the geographical location of enhanced subtropical ridge activity (Sousa et al., 2021). Moreover, blocking in the northern hemisphere is also observed over the Pacific basin - over the West Pacific as well as over the East Pacific. In comparison to the Atlantic and European counterparts being more common in the period from winter to spring, Pacific blocks are most frequent in spring (Barriopedro et al., 2006).

## 2.2   Relevant mechanisms for blocking formation, maintenance and decay

A variety of mechanisms have been linked with blocking, and the balance of mechanisms differs for blocking systems of different types and regions. The interaction of Rossby waves of different scales is a common feature of many mechanisms (Nakamura et al., 1997), often leading to wave breaking and irreversible deformation of potential vorticity contours (Hoskins et al., 1985; Altenhoff et al., 2008). Blocking systems often occurs in regions of weak or diffluent flow which have a lower capacity for wave propagation (Gabriel and Peters, 2008; Nakamura and Huang, 2018), and these regions are often modulated by stationary waves forced by thermal contrasts and continental elevation (Tung and Lindzen, 1979; Austin, 1980).

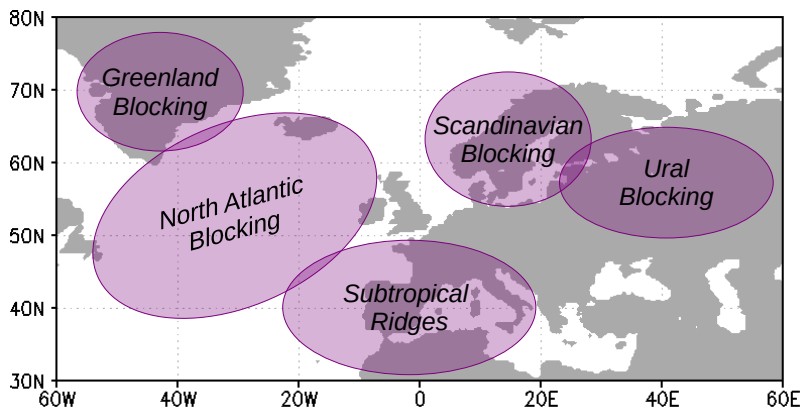

**Figure 1.** Regions over the Euro-Atlantic sector where blocking frequently occurs.

The blocking anticyclone comprises a broad, uniform area of low PV air which has often been advected poleward in the upper troposphere (Crum and Stevens, 1988). Latent heating can also contribute to this formation by enhancing the flow of lower tropospheric, low PV air upwards along warm conveyor belts and into the upper anticyclone (Madonna et al., 2014; Methven, 2015; Pfahl et al., 2015). This is particularly common in blocking systems forming within or just downstream of the oceanic storm tracks (Steinfeld and Pfahl, 2019). Strong cyclone activity in the region upstream is also known to contribute to blocking formation through adiabatic processes as well (Colucci, 1985).

The low PV airmass can be supported by exchange processes between the blocking system and transient eddies, i.e. fast-moving short-lived synoptic-scale systems (Berggren et al., 1949). This can involve a complete replacement of the original airmass by a subsequent wave breaking event (Hoskins et al., 1985) or a subtler "drip-feeding" of low-PV air (Shutts, 1983). While the importance of transient eddy feedbacks seems clear, the precise mechanisms supporting this are still debated (Wang and Kuang, 2019), and the feedbacks maintaining the displaced jet may be important as well as those acting on the blocking anomaly itself (Berckmans et al., 2013).

The mechanisms involved in a blocking system can relate in some cases directly to its impacts. For example, storm activity upstream of the blocking system can lead to high wind and precipitation impacts there (Lenggenhager and Martius, 2020), while in other cases amplified planetary waves can be associated with simultaneous impacts in remote regions (Kornhuber et al., 2020).





## 2.3 Predictability

Blocking is often considered a challenge for prediction systems, but this is only true in some regards. Firstly, blocking can be associated with hemispheric-scale teleconnections, often with influences detected in the tropics (Hoskins and Sardeshmukh, 1987; Moore et al., 2010; Henderson et al., 2016; Gollan and Greatbatch, 2017; Drouard and Woollings, 2018; Parker et al., 2018). At least for these events the intrinsic predictability of the physical system may be relatively high, although biases in models can hinder the realisation of this potential, for example by mis-representing tropospheric waveguides (O'Reilly et al., 2018; Li et al., 2020).

The representation of blocking itself by numerical models has been a longstanding concern (Tibaldi and Molteni, 1990; Pelly and Hoskins, 2003b). Considerable improvement has been made as models have developed (Davini and D'Andrea, 2020), partly through improved resolution (Schiemann et al., 2017) but also through improvements to numerical schemes (Martínez-Alvarado et al., 2018). An overview focused on climate models is provided by Woollings et al. (2018). While many models continue to exhibit serious biases in blocking, it is becoming apparent that only over Europe do models systematically underestimate blocking (Patterson et al., 2019; Davini and D'Andrea, 2020), highlighting the importance of the northern stationary waves and/or specific local processes.

Recent efforts to archive forecasts of weather prediction systems, or in some cases re-forecasts, have shown that blocking remains a challenge on the medium-range weather timescale. In many cases the forecast errors are larger for European blocking compared to other regimes, particularly during the transition phases into or out of a blocking regime (Hamill and Kiladis, 2014). Conversely, during the maintenance phase of a blocking system the errors are often smaller, although the persistence of blocking systems can still be underestimated (Ferranti et al., 2015). Blocking forecast errors remain a concern but, for perspective, the contrast to other regimes is often subtle and requires a large sample of forecasts for statistical significance (Matsueda and Palmer, 2018). While there is room for further improvement, blocking systems are often successfully predicted and this can provide early warnings of related extreme weather.

Several recent studies give specific examples of physical processes which can be improved in models to enable better prediction of blocking. These primarily focus on diabatic processes upstream of blocking systems (Rodwell et al., 2013; Quandt et al., 2019; Maddison et al., 2020), and hemispheric Rossby wave teleconnections, often to tropical structures such as the Madden-Julian Oscillation (Hamill and Kiladis, 2014; Parker et al., 2018; Quandt et al., 2019).

The frequent connection of blocking to hemispheric, and particularly tropical dynamics, provides an opportunity for skillful predictions of blocking variability on monthly, seasonal and even interannual timescales, which is just starting to be realised (Athanasiadis et al., 2014, 2020). Hence, blocking processes could contribute to skillful predictions of related impacts on these timescales, although such predictions would be inherently probabilistic forecasts of, for example, seasonal risk of heat waves or floods.





## 2.4 Impact on Surface Extremes

The strong interest in blocking and its predictability is related to the occurrence of associated high impact weather (Matsueda, 2009). To be more precise, blocking is mainly associated with temperature (Quandt et al., 2017) as well as hydrological extremes (Lenggenhager et al., 2019). Blocking has also associated with other extremes such as marine heat waves (Rodrigues et al., 2019), episodes of low air quality (Pope et al., 2016; Webber et al., 2017) and with wind extremes to a lesser extent.

As reviewed below, the impacts of blocking can vary considerably between seasons and regions, but many impacts arise from one characteristic: the persistence of blocking systems. This persistence is a hallmark of blocking and arises from the dynamics of low-frequency waves, irreversible wave breaking and eddy feedbacks (Hoskins et al., 1985; Pelly and Hoskins, 2003a; Nakamura and Huang, 2018; Drouard et al., 2021). It leads to impact by enforcing prolonged periods of anomalous weather under which surface temperature and rainfall anomalies can build. An exception is the stalling of cyclones upstream of a blocking system (as observed e.g. in winter 2013/2014 over Great Britain); a process which does not necessarily require the blocking system to be persistent.

To give indicative numbers, an average blocking system might last 7-10 days and the most extreme events for 2-3 weeks, though there are significant differences between the statistics of different metrics. Also Drouard and Woollings (2018) argued that recurrence of blocking was more important than persistence *per se* in some cases, such as the Russian heat wave of 2010. They suggested that a seasonal count of the occurrence of blocking was a useful metric for quantifying such impacts.

## 3 Temperature Extremes

### 3.1 Overview

Heat waves and cold spells are long-lasting periods of unusually high or low temperatures, respectively (Robinson, 2001). The extremely high temperatures during heat waves lead to heat stress and thus, a reduction in human comfort (Koppe et al., 2004; Robine et al., 2008; Perkins, 2015). In addition, they increase the risk of heat-related illnesses and mortality (Gasparrini et al., 2015). Also cold spells can cause substantial health risks. Based on data from 13 countries, it was found that 7 % of the total mortality between 1985 and 2002 was due to extreme coldness (Gasparrini et al., 2015). Besides this life-threatening risk, cold spells also influence everyday life by affecting, power supply or public transport. Moreover, both heat waves and cold spells often occur in parallel with droughts, which are periods without precipitation and can cause additional and amplified impacts (see Sec. 4) (Schumacher et al., 2019; Sousa et al., 2020).

Heat waves in Europe are typically associated with co-located anticyclonic circulation features in the free troposphere and high pressure anomalies throughout the troposphere and at the surface (Meehl and Tebaldi, 2004; Cassou et al., 2005; Stefanon et al., 2012; Tomczyk and Bednorz, 2016; Zschenderlein et al., 2019). Accordingly, blocking, which is characterized by persistent anticyclonic flow anomalies, strongly correlates with the occurrence of European temperature extremes in summer. More than 50 % of the most extreme (above the $99^{th}$ percentile) six-hourly maximum temperature events in many regions



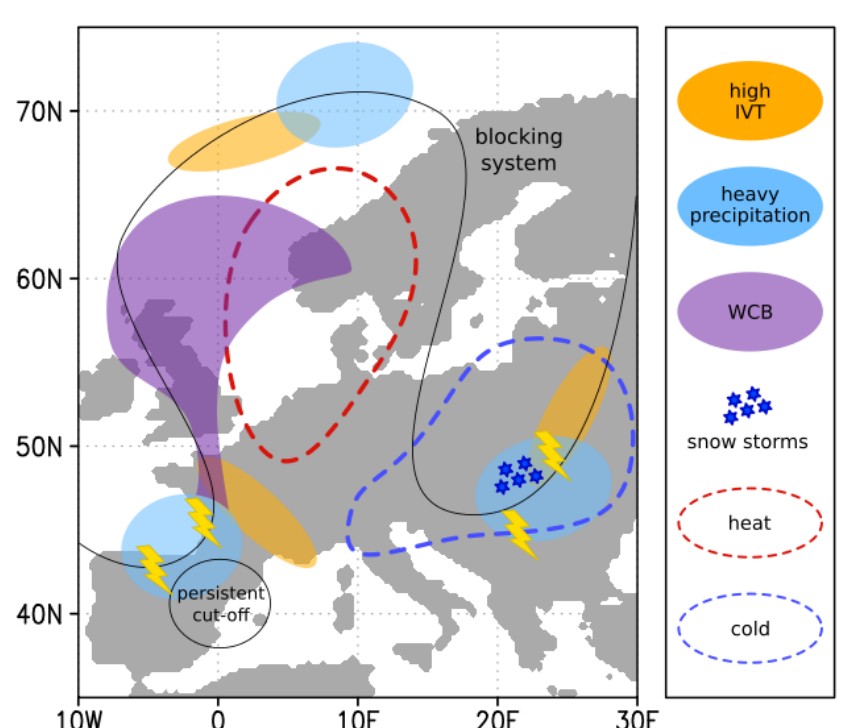

**Figure 2.** Schematic illustration of a blocking system (black line) and some associated surface extremes. Rossby wave breaking occurs on the flanks of the block, leading to (persistent) cut-off systems in this area. Blue stars show areas where snowstorms are observed (eastern flank of the block). Areas with heavy precipitation are marked in light blue (top of the ridge and at both flanks). Areas with high integrated water vapour transport (IVT) are illustrated in orange. Thunderstorm activity is marked by yellow lightnings. The position of a warm conveyor belt appears in purple. Areas where temperature extremes are marked with dashed lines (red for heat waves, blue for cold spells).

in Central and Eastern Europe and more than 80 % in parts of Scandinavia and Scotland have been shown to co-occur with blocking (defined in terms of quasi-stationary potential vorticity anomalies) (Pfahl and Wernli, 2012). In Southern Europe, heat waves typically occur in association with extended subtropical ridges (Sousa et al., 2018), which often do not lead to the overturning of geopotential contours and flow reversal that characterizes classical blocking patterns, but may still be linked to persistent potential vorticity anomalies further north (Pfahl, 2014). Similar to other properties of blocking, the association with heat waves thus depends on the blocking index: anomaly-based indices tend to show stronger correlations with heat waves than blocking indices solely based on flow reversal or wave breaking (Chan et al., 2019).

European cold spells are associated with mid- and high-latitude blocking over the North Atlantic as well as over the European continent. However, in the most cases, the cold anomaly is not located directly below the blocking anticyclone but





downstream or south of it. Over the North Atlantic, blocking is strongly correlated to the negative phase of the NAO that itself is associated with the development of European cold spells. The synoptic pattern during NAO− provides diffluent flow

conditions which are favorable for the onset and maintenance of blocking systems (Luo et al., 2015). However, in general, it is difficult to consider the North Atlantic blocking and NAO− separately form each other as the flow configuration during NAO− itself can be defined as blocking pattern. This results in the development of negative NAO index values during North Atlantic blocking episodes (Croci-Maspoli et al., 2007). The frequency of winter cold anomalies over Europe depends on the exact location of the blocking system (Brunner et al., 2018; Sillmann et al., 2011): The frequency is increased across most of

Europe for blocking over Greenland, while the influence is largest over Central Europe for blocking over the North Atlantic (the influence is larger for systems closer to the continent) and Scandinavia. However, the same blocking system may favor cold anomalies at different locations across Europe (Pfahl, 2014). In numbers, up to 70 % of winter cold spells in Central Europe can be associated with a blocking system anywhere between 60° W and 30° E (Brunner et al., 2018).

## 225 3.2 Dynamics

European heat waves are created by two main processes: heat accumulation due to atmospheric transport and diabatic heating via radiation and surface fluxes (Miralles et al., 2014). Blocking can be conductive to both of these processes, which explains its strong connection to heat waves (Pfahl and Wernli, 2012; Sousa et al., 2018). Although blocking formation itself is often connected to the northward advection of subtropical air masses in the middle and upper troposphere (Nakamura et al., 1997),

recent Lagrangian studies (Bieli et al., 2015; Santos et al., 2015; Zschenderlein et al., 2019) have shown that horizontal advection from lower latitudes is of only secondary importance for the near-surface air in regions affected by heat waves. Rather, the accumulation of heat near the surface is due to descent and adiabatic warming within the blocking anticyclones or subtropical ridges. In addition, this descent is also related to clear-sky conditions that favor surface heating by solar radiation and further diabatic heating of the near-surface air through amplified sensible heat fluxes. This diabatic heating can be further enhanced

by a feedback mechanism with soil moisture (Fischer et al., 2007; Miralles et al., 2019): due to the lack of precipitation in the blocking region (see also Sec. 4), soil moisture is depleted and a larger fraction of the surface-atmosphere heat flux occurs in the form of sensible (in contrast to latent) heat. Soil-moisture feedback and atmospheric heat transport can also act in concert when sensible heat is advected towards heat wave areas from upstream regions affected by drought (Schumacher et al., 2019) and potentially also through feedbacks of the altered surface fluxes on the atmospheric circulation (Merrifield et al., 2019). The

physical mechanisms through which blocking influences heat waves can be amplified due to the persistence of blocking. The lifetime of heat waves increases when they co-occur with a blocking system (Röthlisberger and Martius, 2019), favoring the long-term accumulation of heat.

Cold spells can be favored *downstream* of a blocking system by the horizontal advection of cold air from higher latitudes

or cold land masses (Arctic and Russia) (Bieli et al., 2015; Demirtaş, 2017; Sousa et al., 2018). The transport of the cold air (originated e.g. in the Arctic region) to the target area is characterized by a temperature increase due to adiabatic compression





and turbulent mixing with warmer air masses (Bieli et al., 2015). In addition, blocking systems occurring over the northern North Atlantic can trigger the equatorward displacement of the North Atlantic storm track. The shift of the storm track results in a more southward passage of cyclones towards Europe (Pfahl, 2014). In regions north of these cyclones, cold spells can evolve due to the advection of cold air masses from northeastern and eastern areas (being important for extreme events in Southeastern Europe). Furthermore, the development of cold anomalies in winter can be modulated by persistent clear-sky conditions associated with a blocking anticyclone (Trigo et al., 2004; Demirtaş, 2017). The cloudless sky leads to a strong cooling due to outgoing longwave radiation during nights (diabatic cooling). This process is relevant directly below the blocking anticyclone, thus, it is an *in situ* process. However, there is a temperature increase associated with adiabatic heating due to subsidence in the area of the blocking anticyclone, which may counteract the diabatic cooling (Sousa et al., 2018). Comparing these mechanisms to each other, it was found that the advection of cold air from north and north-east is most important for the evolution of European cold spells (Trigo et al., 2004; Pfahl, 2014; Bieli et al., 2015; Sousa et al., 2018). Cold spells need some time to evolve during blocking situations (Buehler et al., 2011), making the development of a cold anomaly more probable during long-lasting blocking events.

As blocking anticyclones are typically embedded in larger-scale Rossby waves, the relationship between temperature extremes and blocking also translates into a linkage of heat and cold spells to Rossby wave activity. European heat extremes often occur during periods of regionally enhanced Rossby wave activity over the Eurasian continent, while cold spells in Western Europe and parts of the Mediterranean are more associated with enhanced Rossby wave activity over the North Atlantic (Röthlisberger et al., 2016; Fragkoulidis et al., 2018). The persistence of hot spells (cold spells) can be increased (decreased) due to recurrent Rossby wave patterns that amplify in the same geographical region (Röthlisberger et al., 2019). Quasi-resonance of hemispheric wave activity (Petoukhov et al., 2013) may lead to simultaneous heat waves in several regions (Kornhuber et al., 2020). Finally, as for other blocking systems (see Sec. 2), the dynamics of anticyclones associated with European temperature heat waves can be affected by latent heat release in ascending air masses embedded in upstream wave packets (Zschenderlein et al., 2020).

### 3.3 Case Studies

### 3.3.1 Heat Waves

In order to highlight the case-to-case variability in the dynamical processes leading to European heat waves and the role of blocking, we briefly discuss several historical case studies. The most prominent and most severe European heat waves occurred in summer 2003 in Western and Central Europe (Black et al., 2004; Fink et al., 2004; Schär et al., 2004) and in summer 2010 in Eastern Europe (Fig. 3 (a)) (Barriopedro et al., 2011; Grumm, 2011). In 2003, record-high temperatures were measured in June and August, and both months were dominated by anticyclonic circulation regimes (Fink et al., 2004). While the monthly-mean circulation in June was characterized by a broad ridge centered over Central Europe, clear indications of blocking were





observed particularly for the first half of August (Black et al., 2004). Warm air advection may have played a role for the earlier
phases of the heat wave in June (Fink et al., 2004), but during the blocking period in August the flow over France (in the center
of the blocking system) was dominated by stagnant air masses recirculating and descending within the blocking anticyclone
(Black et al., 2004). Positive anomalies in outgoing longwave and incoming shortwave radiation associated with clear-sky con-
ditions point to an important role of radiative forcing, and the precipitation deficit also linked to the predominantly anticyclonic

weather regimes that started already in April led to a positive soil-moisture feedback that strongly amplified the heat wave
(Black et al., 2004; Fink et al., 2004; Fischer et al., 2007; Miralles et al., 2014). The 2010 heat wave mainly affected Eastern
Europe and Western Russia. It was associated with a strong anticyclonic circulation anomaly in the same region (Grumm,
2011) and a reversal of the meridional geopotential height gradient at 500 hPa characteristic for blocking during most of the
period between late June and early August 2010 (Lau and Kim, 2012; Schneidereit et al., 2012). Moreover, the event has been

characterized by a clear positive anomaly in the frequency of subtropical ridges and Omega-type blocks in this longitudinal
sector (Sousa et al., 2021). This blocking anomaly was unprecedented in particular in the eastern part of the region around
50° E. In addition, it was linked to a quasi-stationary Rossby wave train over the Euro-Atlantic sector and more generally,
over the northern hemisphere, consistent with a La Niña teleconnection (Trenberth and Fasullo, 2012; Drouard and Woollings,
2018). Also for this 2010 heat wave, desiccating soils and enhanced surface sensible heat fluxes played an important role (Lau

and Kim, 2012; Miralles et al., 2014; Hauser et al., 2016).

In addition to these most prominent cases, blocking and extended ridges also played a role for other European heat waves.
For example, a heat wave in Western Europe in July 1976 occurred during blocked conditions (Green, 1977). While in summer
2013, a heat wave with its center over Austria and Slovenia was associated with a subtropical ridge extending northeastward

from the Western Mediterranean (Lhotka and Kysely, 2015). A late-summer heat wave in Western and Central Europe in 2016
was mainly driven by subsidence and diabatic heating in the boundary layer within positive geopotential anomalies that were
embedded in eastward-propagating Rossby wave packets (Zschenderlein et al., 2018). In July 2018, Scandinavian blocking
was associated with heat waves over Scandinavia as well as Northern Germany and France (Spensberger et al., 2020). These
cases also illustrate that the location of the high pressure anomaly largely determines which region is affected by a heat wave.


### 3.3.2 Cold Spells

In February 2012, large parts of Europe were affected by extremely low surface temperatures (in many regions 10 °C below
average, Fig. 3 (b)) accompanied by heavy snowfall (de Vries et al., 2013; Demirtaş, 2017). Even Southern European countries
as Italy experienced minimum temperatures of $-15$ °C (WMO, 2015). The cold period affected the traffic sector, health as well

as agriculture. One example for the latter is the damage on grapes in some parts of the growing regions of Moldavia (Planchon
et al., 2015). The occurrence of cold anomalies across Europe was triggered by a persistent ridge-trough-ridge pattern (Demir-
taş, 2017). Both ridges were blocking systems, one amplified northeast-southwest tilted ridge over the Atlantic and one omega
blocking high over Siberia. These upstream and downstream ridges favored the persistence of the trough in between (Demirtaş,



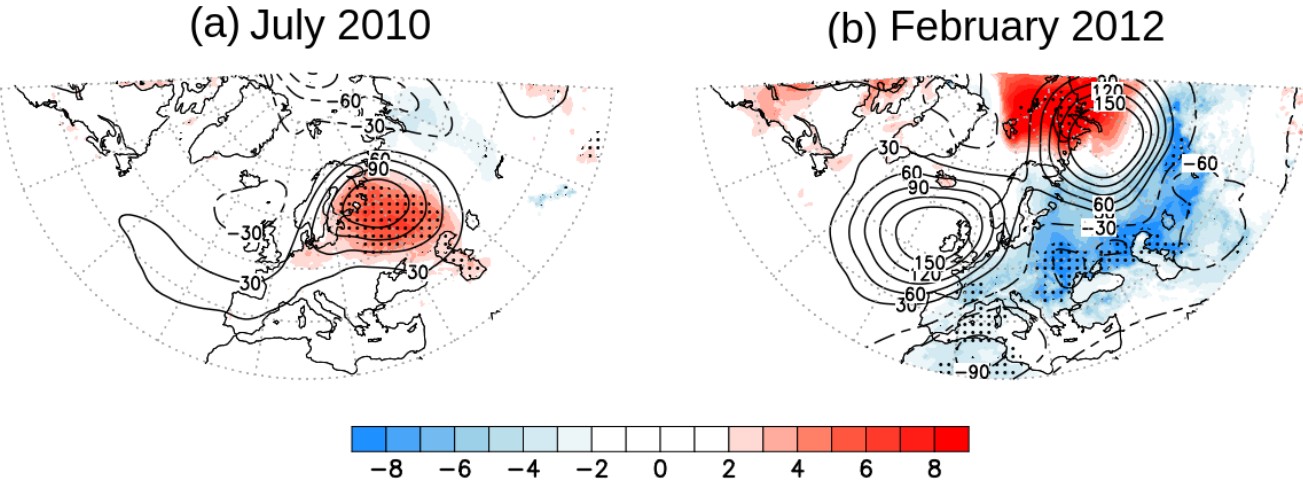

**Figure 3.** Monthly 2 m temperature (in K) and 500 hPa geopotential anomalies (in m$^2$ s$^{-2}$) based on ERA5 data (Hersbach et al., 2020) (a) in July 2010 (in association with the 2010 heat wave) and (b) in February 2012 (in association with the 2012 cold spell) are shown. Dots mark areas exceeding the 2-$\sigma$ level of the climatology (1991-2010).

2017), and thus, the continuous advection of cold air from northern regions. In 2017, the synoptic pattern over Europe was

similar to 2012, with an extension of the Siberian anticyclone towards Scandinavia that blocked the zonal flow and triggered the advection of cold air from the north (Anagnostopoulou et al., 2017). Compared to 2012, the air masses in January 2017 came from much higher latitudes (Anagnostopoulou et al., 2017). These cold air masses favored the evolution of a cold episode over the Balkan Peninsula which was extreme due to its magnitude and to its long duration (Anagnostopoulou et al., 2017). On the other hand, the European cold spell in March 2018 was primarily triggered by the negative phase of the NAO which

seemed to be preconditioned by a sudden stratospheric warming event in mid-February (Karpechko et al., 2018). Although the negative NAO was the dominating flow feature in that case, the extension of the cold spell was also influenced by Scandinavian blocking. At the end of February, shortly before the shift of the NAO from its positive to its negative phase, a blocking system over Scandinavia evolved, pushing the cold air to areas further to the southwest (Ferranti et al., 2019; Kautz* et al., 2020). Similar to the 2018 case, three cold outbreaks in Western and Northern Europe between the end of December 2009 and Mid-

January 2010 were also associated with an extremely persistent negative NAO phase (Cattiaux et al., 2010; Seager et al., 2010; Wang et al., 2010). The negative NAO favored northerly surface wind anomalies leading to a southward advection of cold Arctic air (Wang et al., 2010). The low temperature anomalies coincided with precipitation anomalies which caused an unusual persistence of snow cover (Seager et al., 2010). In addition, this winter has the 2nd highest frequency of North Atlantic blockings since 1949, which is related to the negative NAO phase (being in general favorable for the development of blocking over

the North Atlantic) (Cattiaux et al., 2010). Independently, the low temperatures of this winter were mainly connected to NAO− and less to e.g. Scandinavian blocking (Cattiaux et al., 2010), which was relevant in the 2018 cold spell. However, there is no





study which investigates the role of certain blocking systems for the 2009/2010 winter in detail.

In winter 1941/42 stationary troughs over Europe brought low temperatures and shifted storms tracks affecting the war (Lejenäs, 1989). Another example for an extreme cold event associated with the occurrence of a blocking system and a cy-
clone (blizzard) was observed in March 1987 (Tayanc et al., 1998). Since this event was accompanied by heavy snowfall, it is described in Sec. 4.

## 4 Hydrological Extremes

### 4.1 Overview

In an early blocking study Rex (1950) described the effects of six cases of blocking on the sub-seasonal precipitation distribution over Europe. Conditions were anomalously dry underneath the blocking anticyclones and anomalously wet along the western and eastern flank of the blocking over the Western and Central Mediterranean and the west coast of Scandinavia. Rex attributed these precipitation anomalies to the blocking modulating the location of storm tracks in the blocked longitudes. Namias (1964) and Trigo et al. (2004) confirmed the strong link between blocking occurrence over Europe and precipitation
anomalies using multi-annual data sets. A strong dependence of the location of precipitation anomalies over Europe on blocked longitude exists (Yao and H.i, 2014; Sousa et al., 2017). How the link between blocking and precipitation anomalies translates to hydrological extremes such as heavy precipitation, droughts and floods is discussed in this section.

Droughts have a negative influence on water quality and quantity; this is of major interest since water is vital for diverse
socio-economic activities and for ecosystems. During droughts, the amount of available water is reduced and the water quality is for example impaired by heavy metals or major irons (Zwolsman and Van Bokhoven, 2007). Amongst many impacts of water scarcity, water deficits can lead to crop failure having devastating effects for agriculture (Masih et al., 2014). The lack of water associated with emptier water reservoirs during both summer and winter droughts also influences power generation negatively (Pfister et al., 2006). In summer, dry conditions may also be favorable for wildfires, posing a severe hazard to forests,
cultivated fields or even inhabited areas (Haines, 1989). Another crucial factor is the trapping of gases and aerosols, leading to massive air pollution, which has an impact on human health as well as on boundary layer dynamics and radiation (Finlay et al., 2012; Péré et al., 2013; Athanasopoulou et al., 2014). Furthermore, prolonged dry spells have been shown to enhance soil-atmosphere feedback processes, leading to amplified summer heat waves (see Sec. 3) (Miralles et al., 2014; Schumacher et al., 2019).


Due to its relatively wide latitudinal extent, Europe experiences diverse impacts from blocking, in terms of drought occurrence. Mid- and high-latitude blocking systems have been shown to severely reduce precipitation in the regions directly under the influence of the high pressure system (Sousa et al., 2017). Also, the role of low-latitude blocking and/or subtropical ridges has been recently discussed, showing that these lower-latitude high pressure systems are the main drivers of water scarcity





in Southern Europe (Sousa et al., 2017; Santos et al., 2009). The impact of blocking and ridge episodes in terms of water availability and drought intensity varies with their season of occurrence. For example, in central and northern parts of Europe, a more even distribution of rainfall throughout the year leads to similar impacts of blocking in different seasons, thus severe drought occurrence is dependent on very prolonged large-scale anomalies imposed by blocking systems. However, many European regions' water availability relies on more concentrated precipitation seasons, thus being more susceptible to drought in the case of shorter blocking events coinciding with their wet season. This is notable for example in the Iberian Peninsula, where annual rainfall totals are highly dependent on extended winter rainfall (October to March), or in Eastern Europe, where more substantial summer rainfall constitutes a significant part of annual totals.

Floods are one of the most disastrous weather-related hazards in Central Europe, being on the top of the list of the highest economic losses (Alfieri et al., 2018). Flood events and heavy precipitation (including extreme snowfall) can result in casualties, in high economic losses, and in substantial damages to housing, infrastructure and transport. Long-lasting precipitation periods, serial clustering of heavy precipitation events, or very intensive (convective) rainfall events can all trigger floods. In addition to precipitation, soil moisture content and snow melt may play important roles as hydrological precursors to floods (Merz and Bloschl, 2003). Blocking can influence all of these factors. However, the focus of this section is on the link between blocking, heavy precipitation, extreme snowfall and floods.

Blocking affects regional-scale heavy precipitation in Europe (Lenggenhager and Martius, 2019). Blocking systems change the odds of regional-scale 1-day, 3-day and 5-day heavy precipitation both in summer and winter season. The odds of heavy precipitation events are reduced in the blocked area and high in the areas southwest to southeast of the blocking anticyclone. Often areas of increased odds of heavy precipitation coincide with the tracks of midlatitude cyclones and hence the passage of fronts and warm conveyor belts, hinting at the important role of storm track modulation by the blocking systems (meaning the bifurcation of the storm tracks around the systems) (Sousa et al., 2017; Lenggenhager and Martius, 2019). This is particularly relevant for Southern Europe (including the Mediterranean area), where "classical" blocking configurations can lead to above-average rainfall and extreme precipitation events, opposite to most European areas located further north.

## 4.2 Dynamics

The quasi-stationary nature of blocking imposes persistent large-scale circulation anomalies, dominated by a large area of subsidence and a stable atmosphere in the center of the blocking system. At the same time surface cyclones are guided along the edges of the blocking systems resulting in active storm tracks both to the north and the south of a blocking. These meridional shifts in the storm tracks associated with blocking have been identified as the most general dynamical pattern linking blocking and (heavy) precipitation over Europe (Sousa et al., 2017; Lenggenhager and Martius, 2019). Blocking systems affect the stationarity and pathways of cyclones in their surroundings (Berggren et al., 1949; Nakamura and Wallace, 1989; Swanson, 2002; Booth et al., 2017; Sousa et al., 2017; Nakamura and Huang, 2018; Lenggenhager and Martius, 2019). The lack of cyclones and



the prevailing subsidence occurring in the large area under the blocking center leads to reduced (or even virtually suppressed)
precipitation (Yao and H.i, 2014; Sousa et al., 2017). In this sense, large-scale downward motion is the primary atmospheric
mechanism leading to surface water shortages (and eventually droughts) during persistent blocking episodes.

A strong zonal circulation associated with a positive NAO phase, which reflects more stable stratospheric polar vortex in-
hibit the occurrence of significant Rossby wave breaking (RWB) episodes, thus leading to less favorable conditions for blocking
episodes (Masato et al., 2012), and consequently, to less drought prone conditions in most areas of Europe. However, low lat-
itude structures (in particular subtropical ridges) are frequently the initial stage of an RWB event, as they remain connected
to the subtropical high pressure belt (prior to the occurrence of a cut-off high pressure system, and a subsequent transition to
a mature blocking system). During strong zonal flows, incipient RWB might occur, but generally not leading to mature and
persisting blocking systems. These initial phases of a blocking life-cycle are also important contributors for persistent stable
and dry episodes, in particular in Southwestern Europe, as they tend to block Atlantic frontal activity from reaching the region
(Sousa et al., 2017; Santos et al., 2009). Under these configurations, a tighter jet stream can be found upstream of the ridge,
leading to a significant wet/dry north/south dipole in terms of precipitation anomalies. Under less intense zonal flows, condi-
tions are more favorable for full RWB episodes to develop over the Euro-Atlantic sector, thus leading to more frequent mid-
and high-latitude blocking episodes (Rex type), and thus reverting the aforementioned precipitation anomaly dipole. However,
it has been shown that Omega blocks, which present a subtropical connection (more similar to amplified ridges), tend to pro-
duce higher rainfall deficits in southern Europe, when compared to Rex blocks, where a more intense southerly branch of the
split jet stream can be observed (Sousa et al., 2021).

While these dynamical features can be observed throughout the year, seasonal intricacies exist. For example, during winter,
cold advection near the eastern flank of a blocking system (Sousa et al., 2018; Sillmann et al., 2011) might lead to widespread
surface negative temperature anomalies over the European continent promoting a positive feedback of vertical stability. Also
during the warmer months, positive feedbacks in terms of surface drying can be observed as a consequence of blocking (Fischer
et al., 2007; Seneviratne et al., 2010). Strong radiative fluxes under clear sky conditions (imposed by the blocking structure)
which lead to reduced rainfall and soil moisture (due to evapotranspiration) can lead to severe and detrimental soil desiccation
processes. Previous works have shown that a surface initiated process like this can be amplified and propagate upwards, also
amplifying pressure anomalies, and consequently intensifying a warm core anticyclone (Miralles et al., 2014; Schumacher
et al., 2019).

By affecting the location of cyclones, blocking influence the positions of warm conveyor belts (Grams et al., 2014), fronts,
and atmospheric rivers (Pasquier et al., 2019), which are all important weather features that directly force precipitation. The
detailed mechanisms leading to blocking related heavy precipitation differ regionally and include important interactions with
the local topography, however, there is a general pattern of instability being a key ingredient for blocking related precipitation
in Southern Europe and moisture availability being a key ingredient for blocking related precipitation in Northern Europe



(Sousa et al., 2017).

A high frequency of high-latitude blocking between 0° E and 40° E results in a high frequency of extreme summer precipitation in Romania and the Eastern Mediterranean associated with PV streamers forming on the eastern flank of the blocking systems (Rimbu et al., 2016). Blocking systems over 0° E and 30° E result in a significant increase of the odds of regional-scale heavy precipitation over Greece and the Central Mediterranean (Lenggenhager and Martius, 2019) and reduce the odds of heavy precipitation over Central Europe. Blocking systems over the Central and Eastern Atlantic increase the odds of 1-day heavy

precipitation in summer over several regions in Europe (Lenggenhager and Martius, 2019). This increase is partly linked to a modulation of cyclone tracks by these blocking systems. In addition, blocking also affects the frequency of thunderstorms over Europe during summer (Mohr et al., 2019) and thereby potentially the distribution and frequency of intense convective precipitation (Mohr et al., 2020) and flash floods. Depending on the location blocking systems may both increase or reduce the odds of thunderstorms (Mohr et al., 2019). Blocking over the eastern North Atlantic suppresses the thunderstorm activity over

Central Europe due to northerly advection of colder and more stable air masses into Europe and subsidence. Blocking systems over the Baltic Sea increase the odds of thunderstorms by supporting the advection of warm moist and unstable air masses into Western and Central Europe (Mohr et al., 2019).

Intense and/or long-lasting precipitation may result in flooding. A detailed analysis of the large-scale weather situation during 24 major flood events in Switzerland between 1868 and 2005 revealed that during all summer flood events a blocking system

was present over Russia (Stucki et al., 2012). In the flood events where an upper-level cut-off was central for the generation of the flood triggering heavy precipitation, the cut-offs where slowed down in their eastward propagation by the blocking systems (Stucki et al., 2012). Two detailed analyses of two severe flood cases in Switzerland in October 1868 (Stucki et al., 2018) and in October 2000 (Lenggenhager et al., 2019) confirm the important role of blocking located east of Switzerland for slowing down the eastward propagation of the upper-level PV-streamers and cut-offs that where primarily responsible for the formation

of the heavy precipitation. The stalling of these upper-level structures resulted in very high precipitation accumulations and subsequent flooding.

Blocking-driven high seasonal precipitation accumulations and heavy precipitation in the cold season may translate to substantial snow accumulations if temperatures are cold enough. Blocking contributes to cold temperatures via the advection of cold air along their eastern flank (see Sec. 3). A distinction needs to be made between extreme snowfall on seasonal time-scales and

extreme short-duration snowfall events. Both can be related to blocking. High snow accumulations during entire winters have been linked to anomalous seasonal blocking frequencies. In the winters of 1958 to 1960 several blocking episodes over Europe resulted in negative snow anomalies in parts of Scandinavia (Namias, 1964). The large snow accumulations in Central Europe in winter 2005/2006 were partly related to blocking (Pinto et al., 2007; Croci-Maspoli and Davies, 2009). The Swiss seasonal snow cover is strongly related to Scandinavian blocking (Scherrer and Appenzeller, 2006). Garcia-Herrera and Barriopedro

(2006) report a two-way interaction between seasonal northern hemisphere snow cover and blocking. The proposed 6-step feedback between snow cover and blocking relies on two relationships between blocking and snow cover: first winter blocking over the Atlantic correlates with spring and summer Eurasian and North American snow cover; second spring and summer Eurasian and North American snow cover is associated with an anomalous winter Atlantic blocking activity (Garcia-Herrera





and Barriopedro, 2006). The underlying processes that result in large precipitation accumulations are similar in winter and
summer, in winter blocking in addition affects the partition between liquid and solid precipitation by influencing temperatures
(see section 3). High seasonal accumulations of snow can result in avalanches (Pinto et al., 2007) and may contribute to flood-
ing in the following spring (Stucki et al., 2012).

### 4.3 Case Studies

#### 4.3.1 Droughts

Two prominent examples for droughts in Europe which were associated with the occurrence of blocking were observed in
2003 (Beniston and Diaz, 2004; Cassou et al., 2005; Ogi et al., 2005; García-Herrera et al., 2010) and 2010 (Barriopedro et al.,
2011; Lau and Kim, 2012; Schneidereit et al., 2012). Since these examples are related to mega-heat waves, these cases have
been already discussed in Sec. 3.
Additionally to these mega-heat waves (2003 and 2010) associated with drought in Europe, it is important to highlight other
blocking related droughts in the Mediterranean, a region where model experiments on future climate change scenarios consis-
tently indicate an increased frequency and severity of droughts (Tramblay et al., 2020). One of the most exceptional drought in
the Iberian Peninsula occurred between October 2004 and June 2005 (Fig. 4 (a)). The southern half of Iberia received roughly
40 % of the usual precipitation by June 2005. Moreover, this episode stands out as the driest event in the last 140 years, produc-
485 ing major socioeconomic impacts particularly due to the large decrease in hydroelectricity and agricultural production in both
Iberian countries (Portugal and Spain) (Garcia-Herrera et al., 2007). The blocking activity from October to December within
the Atlantic sector was similar to the corresponding average obtained for the 1958–2005 period, however, the frequency of
blocking events from January to March (the wettest period during the year in the Western Iberian Peninsula) was exceptional,
particularly between 40° and 20° W, where its frequency surpassed the $95^{th}$ percentile of the climatology. Finally, the blocking
activity was remarkable in March, which showed a record-breaking number of blocking days since 1958 while normal blocking
activity resumed throughout spring (Garcia-Herrera et al., 2007).
More recently, between July 2016 to June 2017 a record-breaking drought affected Western and Central Europe, where drought
conditions were observed over 90 % of Central-Western Europe, hitting record-breaking values (with respect to 1979–2017) in
25 % of the area with large socio-economic impacts on water supplies, agriculture, and hydroelectric power production, and
495 leading to devastating forest fires in Portugal (García-Herrera et al., 2019). This dry period was associated with the occurrence
of blocking systems and subtropical ridges which evolved sometimes outside of their typical location (García-Herrera et al.,
2019). Moreover, the occurrence of a mega-heat wave in June 2017 was associated with a long-lasting subtropical ridge which
occurred earlier as expected meaning that such ridges are more likely in July and August (Sánchez-Benítez et al., 2018).





### 4.3.2 Heavy precipitation, Flood Events and Extreme Snowfall

To illustrate the complex interactions between blocking and flood events several flood case studies are discussed in more detail. We start with a thunderstorm case study. A high number of thunderstorms affected Western and Central Europe in May 2018 resulting locally in flash floods (Mohr et al., 2020). During that month a blocking system was located over Northern Europe. The role of the blocking for the series of thunderstorm was twofold. First moist, warm and unstable air masses were advected into Western and Central Europe along the western flank of the blocking system. Second high PV cut-offs repeatedly formed south of the blocking system, they changed the static stability and thereby provided the ideal mesoscale environment for the formation of thunderstorms (Mohr et al., 2020).

We next move to blocking related flood cases in Europe. A rain-on-snow flood event happened in October 2011 in Switzerland. It was associated with strong northerly flow underneath a trough and the subsequent arrival of an atmospheric river resulted in substantial flooding (Piaget et al., 2015). A blocking system over the Atlantic was pivotal both for the amplification of a downstream trough and the transport of high amounts of moisture over the Atlantic (Piaget et al., 2015).

The major floods in Central Europe in June 2013 were associated with high blocking frequencies over Scandinavia and over the Central Atlantic in May 2013 prior to the heavy precipitation event that triggered the floods (Bloschl et al., 2013; Grams et al., 2014). This flow configuration led to cool and wet conditions over Europe prior to the floods. The heavy precipitation episodes that led to the flooding were caused by cyclonic wave breaking on the upstream flank of the Scandinavian blocking and associated equatorward ascending warm conveyor belts (Grams et al., 2014). Similar large-scale flow situations were associated with three other major flood events (2002, 1954, 1899) in the Upper Danube basin (Bloschl et al., 2013). In 2002 blocking over Scandinavia and Eastern Europe slowed the eastward propagation of a cut-off cyclone which resulted in sustained precipitation. In 1954 a blocking system over the Western Atlantic and a cut-off low at its downstream flank were pivotal for the floods.

Blocking played a key role in the formation of a flood event in Southern Switzerland in October 2000 (Lenggenhager et al., 2019). The flood event resulted from a series of three heavy precipitation events over the course of a month with the last event being the most intense and persistent. All heavy precipitation events were associated with upper-level high PV streamers and cut-offs that formed downstream of blocking systems over the Atlantic. The last and most persistent heavy precipitation event was associated with an upper-level cut-off that remained stationary for several days. The cut-off's downstream progression was prevented by a downstream blocking system (Lenggenhager et al., 2019). In October 2000 a feedback between heavy precipitation events could be identified. A large fraction of the diabatically altered low-PV air that reached and strengthened the blocking systems over the Atlantic and Europe was heated in heavy precipitation areas (Lenggenhager et al., 2019).

Finally we discuss high-impact snow events. In March 1987 a blizzard affected the Eastern Mediterranean and the Balkan region (Tayanc et al., 1998). An intense cyclone had formed over the Mediterranean at the eastern flank of a blocking anticyclone located over Europe. This cyclone remained quasi-stationary and in combination with a cold air outbreak along the eastern flank of the blocking system resulted in high-impact snow accumulations (Tayanc et al., 1998). Similarly a snow storm in the Middle East in December 2013 has been linked to Omega-type blocking over Europe with strong anticyclonic wave breaking along the downstream flank of the blocking system (Fig. 4 (b)) (Luo et al., 2015). The anticyclonically tilted trough supported



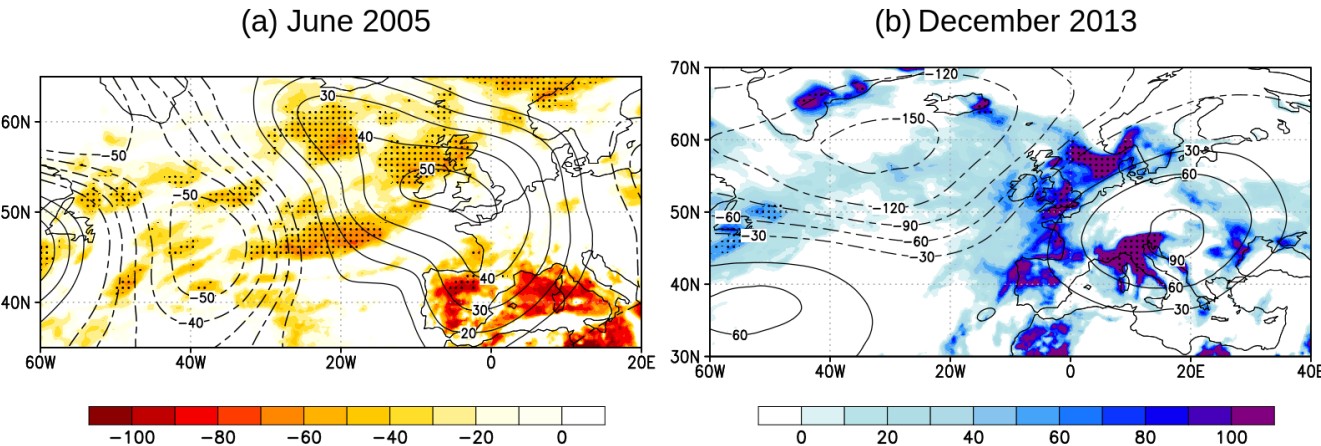

**Figure 4.** Monthly precipitation (in %) and 500 hPa geopotential anomalies (in $m^2\,s^{-2}$) based on ERA5 data (Hersbach et al., 2020) (a) in June 2005 (in association with an Iberian drought) and (b) in December 2013 (in association with the 2013 snow storm in the Middle East) are shown. Dots mark areas exceeding (a) the 1-$\sigma$ or (b) the 2-$\sigma$ level of the climatology (1991-2010).

the snow storm both through cold air advection and through forced lifting (Luo et al., 2015). Record-breaking snowfall hap-
pened the northern part of the Alps in January 2019. The snow event was associated with North Atlantic blocking. A persistent blocking system transported moist air from the North Atlantic towards the Alps (Yessimbet et al., 2021).

## 5 Other Extremes

### 5.1 Wind Extremes and Storm Surges

Additional to its relevance for precipitation and temperature extremes (see Sect. 3 and 4), the presence of a blocking system can also be instrumental for the occurrence of wind extremes. Following Pfahl (2014), this influence can be distinguished in two main variants. First, as discussed in Sec. 3 the presence of a blocking system leads to the deflection of cyclones to the regions surrounding the system, resulting both in areas with high wind extremes and calms (low wind extremes). For example, given a high latitude blocking system over the North Atlantic-European region, the jet stream and the storm track is shifted southward,
corresponding to a negative phase of the NAO (Wanner et al., 2001). A good example of such a large-scale atmospheric set up is the winter 2009/2010, with a very persistent blocking system near Greenland which extended to Scandinavia around 60° (Santos et al., 2013). This led to enhanced cyclone activity extending from the North Atlantic into the Mediterranean area and strong mean wind conditions around 45° N, while low cyclone numbers and reduced mean winds were found near Greenland and Iceland (Santos et al., 2013). In the winter 2011/2012, the opposite situation occurred: a very persistent blocking system
was located over Western Europe, around 50° N, 0° E (Santos et al., 2013). This was associated with a northward shift of the





jet stream, and a very positive NAO phase. In this case, cyclone numbers and mean wind speeds were very low around Western and Central Europe, while the number of cyclones and wind speed was enhanced near Greenland (Santos et al., 2013).

Second, blocking (or more precisely the associated surface high pressure system) can be determinant to establish - together with a low pressure anomaly - a very strong near-surface pressure gradient. A strong pressure gradient implies in turn strong geostrophic near surface winds. A good example of such a situation is storm Kyrill in January 2007, where the presence of an intense blocking system over Southern Europe contributed to an unprecedented high pressure gradient over the German Bight and Baltic Sea before the storm Kyrill crossed the area (Fink et al., 2009). With the passage of the cyclone over the North and Baltic Seas, the pressure gradients increased even further, leading to widespread strong winds and wind gusts over Western and Central Europe and significant impacts (Fink et al., 2009). Based on the ERA-Interim data set and thus a large number of events, Pfahl (2014) confirmed the above insights and provided evidence that both cyclones and blocking are often present during the occurrence of wind extremes in Western and Central Europe. While cyclones are generally located north/northeast of the affected location, blocking is typically found south or southwest of the location, and thus opposite to the location of the cyclones. This set up is more consistent for wind extremes over Central Europe and less consistent for Southern Europe. In summary, Pfahl (2014) provides clear evidence for the importance of surface anticyclones and blocking for the occurrence of wind extremes.

Specifically for extreme wind situations affecting the Iberia Peninsula, Karremann et al. (2016) have defined an "hybrid type" atmospheric set up (Fig. 5), which is characterized by a strong pressure gradient over Iberia due to the juxtaposition of low and high pressure centers in the area. This type corresponds to about 30 % of the cases for wind extremes affecting Iberia. The later study of Hénin et al. (2020) further documented the importance of the "hybrid type" cases for the occurrence of wind extremes in the region, though the relative number of extreme wind events associated with this large scale set up is lower (about 15 %). Differences between both studies are primarily attributed to the different reanalysis data sets and periods analyzed.

Both persistent strong and persistent low wind conditions can have important impacts, for example, related with the energy production. Weber et al. (2019) analyzed the synoptic conditions leading to persistent (high or low) wind power production in the German Bight. Prolonged calms are primarily associated with nearby atmospheric blocking events, which can have a life time of up to several weeks. Still, this is not the case for all low-wind situations, and some variability in the weather patterns is identified. The predominant characteristic is that the longest cases are associated with very weak pressure gradients over the area, and thus calms with low wind speeds and typically high persistence (Weber et al., 2019). This is in line with Grams et al. (2017), who analyzed the relationship between weather regimes and wind energy production in Europe. Regarding persistent strong-wind situations, they are typically associated with westerly flow and north-south strong pressure gradients. A new insight in this study is that high-wind periods may be distinctly longer than low-wind periods. This may be unexpected, as surface cyclones are typically more transient than high pressure centers. However, and for at least one of the analyzed longer lasting periods, the temporal clustering of cyclones (Mailier et al., 2006; Pinto et al., 2014) has apparently played a crucial role in maintaining the strong wind conditions for the extended period.

Costal storm surges are typically associated with the passage of a cyclone (or cyclones) near coastal areas, which push the surface water masses towards the coast through wind stress (e.g. Dangendorf et al., 2016). In synoptic terms, these situations





are often characterized by the presence of a long-lasting large-scale pressure gradient, which is then strengthened by the passing storm itself, thus leading to sustained winds and, in combination with the tides, to a high water levels in specific coastal areas. Analogously to the examples given above, a closer look reveals a large-scale set up with the juxtaposition of a high and a low pressure center, the former often being associated with blocking. A notable example is the storm surge in the German Bight

in December 2013 which was associated with the passage of storm "Xaver" (Dangendorf et al., 2016). While storm "Xaver" passed over Southern Scandinavia in an E-SE path, the presence of a strong anticyclone over Southwestern Europe created an impressive and long enduring pressure gradient over the German Bight (Dangendorf et al., 2016; Staneva et al., 2016). This led in turn to sustained north-northwesterly winds, resulting in the extraordinary storm surge characterized by record breaking sea levels in several coastal stretches in Lower Saxony (Jensen et al., 2015; Dangendorf et al., 2016). The presence (or not)

of a long-lasting pressure gradient is crucial given that it is the combination/interaction of the surge and the tidal components (Horsburgh and Wilson, 2007) that leads to the actual coastal high waters – the longer the sustained large-scale winds (and thus the surge), the more probable high coastal water levels may be. Janjic et al. (2018) analyzed a number of storm surges affecting Ireland in the unprecedented storm winter of 2013/2014 (Matthews et al., 2014). Several of the analyzed cases (including the above described storm "Xaver") point to synoptic situations with a juxtaposition of a passing low pressure center close to

Scotland and the presence of and a blocking system or an anticyclonic ridge to the south, thus inducting a strong pressure gradient. Focusing on the last 100 years, Haigh et al. (2016) investigated storm surges across the UK coasts. They stress the role of temporal cyclone clustering (Mailier et al., 2006; Pinto et al., 2014) for the longer surge events. Given the typical dynamical set-up associated with cyclone clustering, it is assumed that high pressure systems associated with long-lasting blocking systems at lower latitudes have contributed to the strong impacts in these situations.

## 5.2 Compound Events

Compound events are defined as a "combination of multiple drivers and/or hazards that contribute to societal or environmental risk" (Zscheischler et al., 2018), according to the IPCC SREX (Leonard et al., 2014) and within the IPCC risk framework. In Zscheischler et al. (2020), a typology of compound events was proposed and analytical and modelling approaches were suggested to aid in the investigation of compound events. The four classes defined in Zscheischler et al. (2020) are:

– preconditioned, where a weather- driven or climate- driven precondition aggravates the impacts of a hazard;

 – multivariate, where multiple drivers and/or hazards lead to an impact;

 – temporally compounding, where a succession of hazards leads to an impact;

 – and spatially compounding, where hazards in multiple connected locations cause an aggregated impact.

Preconditioned compound events can also be connected to blocking activity. The example mentioned in Sec. 4 - a rain-on-snow

event in October 2011 in Northern Switzerland - was associated with strong northerly flow underneath a trough and the subsequent arrival of an atmospheric river, which resulted in substantial flooding in the region. A blocking pattern over the Atlantic was responsible for both the amplification of the downstream trough, as well as the transport of high amounts of moisture over

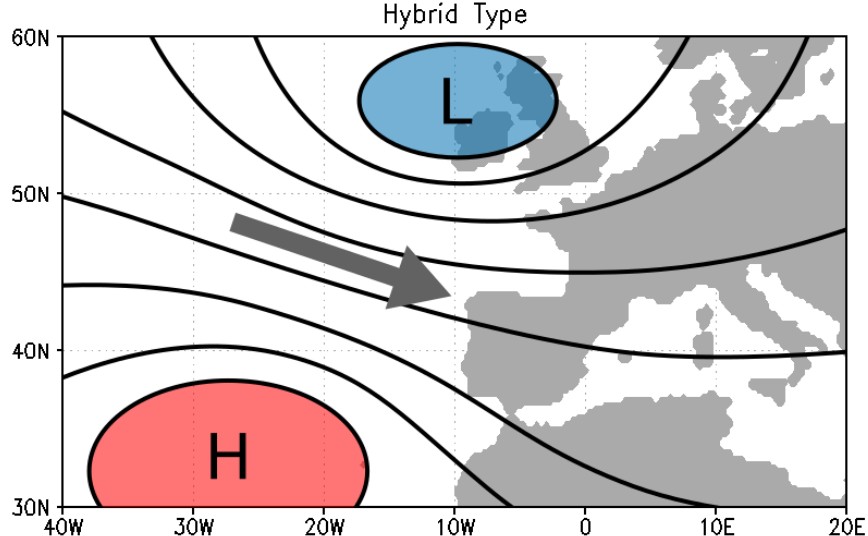

**Figure 5.** Schematic illustration of the hybrid type pattern with a low pressure system over the British Isles and a high pressure system over the southern North Atlantic (Hénin et al., 2020).

the Atlantic (Piaget et al., 2015). Indeed, most of the case studies which have been presented so far fall within the definition of compound events, being good examples of multivariate events. For instance, concurrent droughts and heat waves can occur on
different timescales. On shorter timescales, compound hot and dry conditions are attributable to blocking (Röthlisberger and Martius, 2019), and to soil moisture–atmosphere interactions as mentioned in the different case studies (Fischer et al., 2007; Miralles et al., 2014). On longer timescales, compound dry winters/springs and hot summers seem to be occurring more often especially in the Mediterranean region, due to land–atmosphere interactions (Schumacher et al., 2019). Concurrent droughts and heat waves can cause additional and amplified impacts (e.g., wildfires, crops losses, natural vegetation death, power plants,
reduction of fisheries) (Zscheischler et al., 2020). Another important class of compound events that can be related with blocking patterns are temporally clustering events. In Subsec. 5.1, it was mentioned a case of temporal cyclone clustering (Mailier et al., 2006; Pinto et al., 2014) that had impact on longer storm surges or even extreme precipitation (Priestley et al., 2017). Given the typical dynamical set up associated with cyclone clustering, it is assumed that high pressure systems associated with long-lasting blocking systems at lower latitudes have contributed to the strong impacts in these cases studies (see Subsec. 5.1).

## 6   The Impact of Atmospheric Blocking on the Predictability of Extreme Weather Events

The prediction of blocking is still a challenging task due to the complex character of these systems. In particular, this complexity is related to the interactions with other flow features (e.g. transient eddies) on different spatial and temporal scales.





Nevertheless, blocking occurrence can also be a source of predictability in medium-range and subseasonal forecasts (Vitart
et al., 2014; Quandt et al., 2017) which may benefit the predictions of related surface impacts. However, as shown in the previous sections, surface impacts of blocking systems are highly variable, raising the question if there is an increased predictability for surface extremes caused by the occurrence of blocking compared to non-blocking situations.

Predictability of heat waves related to blocking on daily to seasonal time scales may, in principle, arise from the dynamics
ics and persistence of the blocking as well as from soil moisture feedbacks. Blocking formation and maintenance are mainly driven by atmospheric processes (see Sec. 2) with characteristic time scales and thus potential predictability of several days to one-two weeks, but can also be influenced by boundary conditions such as sea surface temperatures and teleconnections offering potential predictability on longer time scales (Trenberth and Fasullo, 2012; O'Reilly et al., 2016; Ferranti et al., 2018). The predictability of blocking is generally higher in its maintenance phase compared to blocking onset and decay (Tibaldi and
Molteni, 1990; Reynolds et al., 2019). Soil moisture conditions are sensitive to precipitation accumulation over the preceding months and may thus yield predictability on seasonal and even multi-year time scales (Quesada et al., 2012; Breil et al., 2019). Altogether, there are indications that these processes related to blocking persistence and soil moisture give rise to a higher subseasonal predictability of heat extremes compared to average summers (Wulff and Domeisen, 2019). The linkage to blocking also provides opportunities for skillful statistical forecasts of heat waves (Chattopadhyay et al., 2020). Detailed studies of the
predictability of the Russian heat wave in 2010 (Matsueda, 2011; Quandt et al., 2017, 2019) showed that the predictability of the blocking systems was generally high, in particular with respect to their onset and maintenance. Lower predictability was associated with the decay of the main blocking system and some details of the blocking characteristics (Quandt et al., 2017). These predictability differences were linked to upstream Rossby wave dynamics and moisture transport (Quandt et al., 2019). Cold spells have a significant predictability within a two-week lead time, however with a strong decrease during the first week
of forecasts and a generally reduced predictability during their onset and end phases (Lavaysse et al., 2019). Ferranti et al. (2018) investigated the impact of large-scale flow patterns and their transitions on the predictability of cold conditions over Europe. In this study, they introduce a novel framework dealing with the transition of (non-)blocking and NAO (+/−) situations (NAO-BLO phase space). They applied their method to reanalysis data and could show that NAO+ favors transitions into blocking, while blocking itself favors transitions into NAO−. They further investigated extended range multi-model en-
semble forecasts and found that the predictability of severe cold events depends on the type of transitions. For example, the extended-range ECMWF ensemble shows increased predictability of cold spells associated with the development of Greenland blocking. The forecast variability of the late winter cold spell in March 2018 (see Sec. 3) was investigated by Kautz* et al. (2020). The analysis of ECMWF sub-seasonal ensemble forecasts could show that the occurrence of a Scandinavian blocking anticyclone at the end of February as well as the regime shift to a strong NAO− phase influenced the predictability of the
cold spell. Ensemble members which showed the NAO− pattern also captured the cold spell. On the other hand members which additionally captured the precursor blocking over Scandinavia featured a better representation of the south-eastern extension of the cold spell. Besides the studies which investigated the relation between cold spell and blocking predictability in operational models, there are also efforts to produce skillful seasonal predictions of winter blocking and temperature extremes



with the help of statistical models. For example Miller and Wang (2019) developed a multiple linear regression model using predictors based on sea surface temperature, 70 hPa geopotential height and sea ice concentration which skillfully predicts the blocking frequency over Eurasia. In addition, they developed similar models also addressing the relationship between blocking and surface temperature extremes. With these models, cold anomalies over Eurasia and Greenland would be skillfully predicted.

While there are a few studies which connect the predictability of temperature extremes on the medium range and the subseasonal time-scale with the occurrence of blocking, there are hardly any comparable studies which focus specifically on the predictability of wind or rain extremes in connection with blocking. An exception are studies that relate the predictability of wind extremes to the NAO (which is strongly connected to blocking, as pointed out in Sec. 3). Scaife et al. (2014) have shown that winter wind predictions over Europe benefit form the influences of the NAO, meaning that strong wind events are poorly predicted in regions, where the NAO influence is weak. However, there is a lack of studies that examine the relationship between the predictability of low and high wind events and blocking directly. This could be a perspective for future research, especially as skillful predictions of wind extremes are highly relevant for the renewable energy sector. At the eastern flank of the blocking system, which was associated with the Russian heat wave in 2010, heavy rain events led to flooding in Pakistan (Lau and Kim, 2012). As mentioned above, the impact of the blocking on the predictability of the heat wave was investigated (Quandt et al., 2017), but not how blocking has influenced the prediction of the extreme rain in Pakistan. There are only studies which deal with the predictability of the Pakistan floods without any linkage to blocking (Webster et al., 2011). Another example is the 2013 flooding event in Europe. Ionita et al. (2015) investigated its predictability by using a multiple linear regression model. They could show that an accurate prediction of the June 2013 Elbe River extreme discharge was possible by considering the amount of precipitation in May and June as well as May soil moisture and sea level pressure. They did not used blocking as predictor for their model. However, they also discussed the synoptic flow pattern and emphasized that the persistent blocking system which evolved in mid-May over Scandinavia could have been already considered as an indicator for a potential flood.

## 7 The Relation between Atmospheric Blocking and Extreme Weather Events in the context of Climate Change

The relevance of extreme weather events for society may increase in the coming years, as regional changes in the magnitude and frequency of these events are expected due to global warming (Mitchell et al., 2006; Rahmstorf and Coumou, 2011; Coumou and Rahmstorf, 2012; Mitchell et al., 2016). Changes in the characteristics of several of these extreme events have already been detected in observations over the last decades (Folland et al., 2002; Sparks and Menzel, 2002; Rahmstorf et al., 2012), and are evident in climate projections for the near and distant future (Kjellstro¨ M et al., 2011; Branković et al., 2012). As blocking systems can trigger weather extremes, it is also of interest how blocking characteristics may change in the future and how these changes can in turn influence the occurrence and characteristics of surface extreme weather events.





Future projections show generally a decrease in blocking frequency over the mid-latitudes, but there are hints for an increase in blocking duration (Sillmann and Croci-Maspoli, 2009). However, changes blocking occurrence cannot be generalized for the entire northern hemisphere, as there are strong regional differences. For example, the frequency and duration of Ural blocking is projected to increase under future climate conditions (Luo et al., 2018). In contrast, the frequency of blocking systems over

the Euro-Atlantic sector shows a significant decrease in climate simulations in future decades, independent of the considered blocking duration (Matsueda et al., 2009). This decrease is focused on the western flank of these blocking systems, whereas an increase is predicted on their eastern flank, indicating an overall shift of blocking activity towards Eurasia (same location where the 2010 Russian heat wave blocking was observed) (Matsueda et al., 2009; Masato et al., 2013b). A shift is expected not only zonally, but also meridionally: A poleward shift of blocking activity in summer indicates that there will be more high-latitude

blocking during this season, but less in the mid-latitudes (Masato et al., 2013b; Matsueda and Endo, 2017). Moreover, the size of blocking systems over the northern hemisphere is projected to increase with climate change (Nabizadeh et al., 2019). The expected changes in blocking depend on several factors, including stratospheric variability, changes in the mid-latitude jet stream with respect to intensity as well as location, and near-surface Arctic warming (Francis and Vavrus, 2015; Kennedy et al., 2016).

The expected future changes in blocking might also influence characteristics of future European heat waves. Climate models project a general increase of extreme summer temperatures and heat wave intensity (Meehl and Tebaldi, 2004; Fischer and Schär, 2010; Perkins et al., 2012) that is mainly related to the thermodynamic effects of climate warming. Under future climate conditions, blocking is projected to remain the most relevant circulation feature initiating European heat waves (Brunner et al., 2018; Schaller et al., 2018). Despite some indications of a weakening of the midlatitude circulation in summer in recent decades

(Coumou et al., 2015; Horton et al., 2015), studies on future changes in the properties of weather systems associated with heat waves (such as blocking persistence) come to diverging conclusions (Plavcova and Kysely, 2013; Brunner et al., 2018; Mann et al., 2018; Schaller et al., 2018; Jézéquel et al., 2020; Huguenin et al., 2020). Whether future changes in blocking dynamics might lead to changes in heat waves beyond their thermodynamic intensification thus remains an important question for future research.

The relation between blocking and low temperature extremes will remain relevant in the future. Under future climate conditions, the presence of long-lasting blocking systems over the North Atlantic increases the probability of low surface temperatures over the European continent in winter (Sillmann et al., 2011). Climate projections show that Western European cold spells will become comparatively warmer and may partly remain above the freezing point under future climate conditions (de Vries et al., 2012). These changes in cold spell characteristics can be partly associated to changes in blocking, whereas changes in other

large-scale patterns (zonal temperature gradient and strength of the westerlies) have an additional contribution. For example, in winter thermal advection leading to cold extremes could weaken as a result of the weakened mean temperature gradients (Kennedy et al., 2016).

Regarding precipitation, negative anomalies (associated with dry conditions) around the British Isles as well as positive anomalies (associated with wet conditions) along the south-eastern coast of Greenland and over parts of the North Atlantic are related

to European blocking (Sillmann and Croci-Maspoli, 2009). Climate projections show that these patterns will increase in a





warming climate. The increase of positive precipitation anomalies over the North Atlantic is related to the passage of cyclones at the southern flank of the blocking systems.

The changes in blocking features (such as frequency) can affect the occurrence of extreme surface weather. However, since
the occurrence of such extreme weather events is also influenced by other factors (e.g. thermodynamical factors), it is not possible to transfer the changes in blocking one-to-one (Woollings et al., 2018; Nabizadeh et al., 2021).

## 8   Outlook and Research Perspectives

In the previous sections we have provided an overview on the relationship between the occurrence of blocking and different
types of extreme events. This said, it is important to note that not every blocking system leads to the occurrence of an extreme weather event, and extreme weather events can also be favored by other large-scale flow patterns (such as an intense zonal flow) (Trigo et al., 2004; Priestley et al., 2017). In particular, the persistence of blocking alone is not necessarily a meaningful indication that an extreme event will actually occur. The spatial component also plays a decisive role, i.e. the actual location of the blocking system is crucial for the formation and type of extreme event. In this review, we have shown that blocking systems
are capable of triggering a variety of extreme events (e.g. heat waves or flooding), sometimes even directly after each other at the same location or at the same time (known under the term *compound events*). Several research gaps have been identified, for which research perspectives are given below.

The **dynamic relationships** between extreme weather events and blocking are generally not fully understood. This is also
due to the fact that the dynamics of blocking systems are complex, in particular because of the partly nonlinear interactions with the large-scale flow as well as with other weather systems (e.g. transient eddies). In addition to the interactions within the troposphere, there are feedback mechanisms with the land masses and the oceans as well as coupling processes with the stratosphere. Especially the latter are not yet completely understood.

The results on (statistically significant) correlations between two phenomena depend on the choice of **detection methods**, which is also true for blockings and weather extremes. Of the approximately 250 studies referenced in our article, about half deal directly with weather extremes associated with blockings, and there are twice as many studies of temperature anomalies than of hydrological events. 42 % of the studies that address weather extremes associated with blockings do not use a blocking index. Instead, most of rely on synoptic descriptions or refer to other studies that have identified the responsible system as
blocking - 40 % of these are case studies. 58 % of the studies that discuss weather extremes associated with blockings use a blocking index - only 28 % of these present case studies, which is probably related to the fact that a detection method is mandatory for analysis of long data sets and to derive statistical correlations. In the studies we looked at, different blocking indices (cf. Sec. 2) were used for both hydrological and temperature extreme events, which shows that the relations between blockings





and these types of extremes can usually be observed independently of the choice of blocking index. Nevertheless, there is a
dependence here, as described in Sec. 3 for heat wave and as investigated in Chan et al. (2019). Therefore, an investigation
of the extent to which the choice of blocking index plays a role in the analysis of different types of extremes is an interesting
question for further research.

Through **field campaigns**, data on blocking systems and/or on associated extremes can typically be collected only over a
small area, but at a high temporal and spatial resolution (e.g. 2016 NAWDEX block case study (Maddison et al., 2019; Stein-
feld et al., 2020) or the 2003 UK TORCH campaign (Vieno et al., 2010)). **Long-term measurements**, both in-situ and from
remote sensing systems, also provide important data as outliers (e.g., temperature records) are captured (Brunner et al., 2017).
The analysis of such data can help to improve our understanding, for example, as input for further simulation studies. With
respect to coupling processes between the troposphere and stratosphere, which can influence blocking and thus also extreme
weather events, there is a lack of such long-term measurements. The potential of satellite data for blocking analysis has been
shown in Brunner et al. (2016) and Brunner and Steiner (2017). They show for high-impact blocking events (e.g the blocking
system associated with the 2010 Russian heat wave) that GPS radio occultation observations can be used to capture the vertical
structure of blocking systems.

A better understanding of the dynamics through more and better observations would also have a positive effect on the cap-
turing of blocking systems and their interactions in **numerical models**. Currently, the representation of blocking in numerical
weather and climate models has several weaknesses, both regarding its frequency, intensity and persistence. If we look explic-
itly at the life cycle of blocking systems, the onset and the decay phases are the periods that pose the primary challenge for
numerical models (Frederiksen et al., 2004). If the blocking system is part of the initial conditions in a weather forecast, this
can have a positive effect on the predictability of surface conditions (also on sub-seasonal time scales) due to the potential
persistence of the blocking system. However, because blocking decay is difficult to predict, the duration of blocking systems
is also often incorrectly predicted. It is precisely the poor predictability of the decay of the blocking system that inevitably
transfers to or at least influences the predictability of the decay of an extreme weather event (Quandt et al., 2017). Neverthe-
less, recent studies have used the relations between blocking and temperature extremes and provided evidence of improved
predictability for the surface conditions (Ferranti et al., 2018). However, there are hardly any comparable studies which focus
specifically on the predictability of wind extremes or heavy precipitation in connection with blocking. Further studies are thus
needed to quantify the potential of these relations for **weather forecasting** and also **climate projections**.

When considering the relationships between blocking systems and extreme events, we are pursuing a **"moving target"** as
both decadal variability and long-term global climate change modulate the occurrence of both blocking and extreme weather
events. We arguably have a very limited sample of observed extreme events relevant to today's climate. The consideration of
larger samples of data (Maher et al., 2019; Ehmele et al., 2020) is thus of key important to provide robust estimates of these
relationships in a changing climate. Whether future changes in blocking dynamics might lead to changes in heat waves beyond





their thermodynamic intensification thus remains an important question for future research. In this line of research, attribution
studies can provide another important piece of the puzzle (Stott et al., 2016; Swain et al., 2020). Therefore, further studies are
needed to understand and estimate the relation between blocking and weather extremes under climate change.

Further studies on blocking and their influence on weather extremes are needed to understand the underlying physical
mechanisms. Both observational data sets, which provide an important data basis, and new modeling strategies can provide a
possible perspective to improve this understanding as well as the predictability and risk assessment of extreme weather events.
Potential changes in weather extremes due to global climate warming also lead to an increase in the need for better forecasting
and risk assessment. For planned observational campaigns as well as long-term observations, feedback and coupling processes
should be covered in particular. The same applies to model experiments, for example realized in seamless modeling approaches.

*Data availability.* ERA5 (Hersbach et al., 2020) is available after registration at https://cds.climate.copernicus.eu/#!/home (last access: 3 Au-
gust 2021)

*Author contributions.* LAK wrote the introduction (Sec. 1), the parts addressing cold spells (Sec. 3), and the sections about predictability,
climate change and research perspectives (Sec. 6-8). TM wrote the background section (Sec. 2). SP wrote the passages on heat waves (Sec. 3)
and contributed to the Sect. 6 and 7. In Sec. 4, OM wrote the parts on heavy precipitation, flood events and extreme snowfall, while AMR
and PMS wrote the parts addressing droughts. In Sect. 5, JGP wrote the parts on wind extremes and AMR on compound events. The figures
were prepared by LAK, with the idea for Fig. 2 coming from OM. All authors contributed with discussions and revisions.

*Competing interests.* SP is executive editor and co-editor of WCD. TW is co-editor of WCD. The other authors declare that they have no
conflict of interest.

*Acknowledgements.* We thank the ECMWF for access to the ERA5 reanalyses. JGP thanks the AXA Research fund for support. PMS
would like to thank project HOLMODRIVE (PTDC/CTA-GEO/29029/2017), funded through FCT (Fundação para a Ciência e a Tec-
nologia, Portugal). AMR acknowledges the Fundação para a Ciência e a Tecnologia, Portugal (FCT) through the project WEx-Atlantic
(PTDC/CTAMET/29233/2017) and Scientific Employment Stimulus 2017 (CEECIND/00027/2017). OM acknowledges support from the
Swiss National Science Foundation Grant No. 178751.





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
