# Peer review of "Atmospheric Blocking and Weather Extremes over the Euro-Atlantic Sector - A Review"

_Weather and Climate Dynamics, 2021_

## Author Response (AR1)

**RC1: Comment on wcd-2021-56 - Anonymous Referee #1**

**https://doi.org/10.5194/wcd-2021-56-RC1, 2021**

The manuscript by Lisa-Ann Kautz and co-authors presents an extensive review of the state of research on atmospheric blocking with a strong focus on their impacts. It starts with an overview of blocking types including the most important mechanisms during a blocking live-cycle. Then the impacts of blocking on several types of extreme events are addressed: temperature, hydrological, wind, and compound. For each extreme an overview is given, the involved dynamics are explored and some case studies are given. In the final part of the manuscript the predictability of blocking induced extremes events as well as their relationship in a changing climate is investigated. The manuscript ends with a summary of the most important open research questions in relation with blocking.

This review manuscript presents a timely and extensive overview of the manifold blocking impacts, that can sometimes seem contradictory at first glance (hot/cold and wet/dry extremes can both be caused by blocking). The topic it addresses is well motivated, it is well-written and -structured. Mostly, the authors manage to generalize and combine results from different studies to clear top-level messages (one exception is mentioned in my comments below). My only real point of critique are the first two figures: they are never mentioned in the text and the information I could extract from them was limited. I think both of them aim to address important topics (blocking locations and their naming as well as impacts depending on their relative position to the block) but fail to fully do so.

Apart from that I only have a few minor comments outlined below. Given that the authors address them my evaluation is that this manuscript should be published in Weather and Climate Dynamics.

Reply: We thank the reviewer for his/her generally positive feedback and are grateful that he/she supports a publication in WCD.

We understand the main criticism addressing the figures, especially Figure 1 and 2, and we agree they could be better explored and interlinked with the text. We have replaced the old Figure 1 by a new figure which defines blocking regions in a clearer way. Moreover, we have added corresponding references and some explanations to the text.

Regarding Figure 2, we aimed to provide a summary of the many influences blockings can have for different types of natural hazards. Therefore, we have opted to add everything into one figure as a synthesis instead of splitting the surface influences into several figures. As some of the processes shown in Figure 2 can occur simultaneously, we think this is the more adequate representation. Specifically, we rather want to show here where the extremes can occur relative to the blocking - using an omega block as the example. Nevertheless, as suggested by Anonymous Referee #2, we divided the figure in two panels – one for summer season and one for winter season, in order to address the seasonal differences. In addition, we have added a description of Figure 2 to the text.

L115-122: *"The different blocking areas dealt with in this study are shown in Fig. 1 (Shabbar et al., 2001; Buehler et al., 2011; Luo et al., 2016; Rohrer et al., 2019; Sousa et al., 2021). These are: Greenland (GL), the North Atlantic (N-ATL), Europe (EUR), the subtropics (SUBTROP) and the Ural Mountains (URAL). The areas shown are partly overlapping. Scandinavian blocking also belongs to the category of European blocking, the southern parts of North Atlantic and European blocking overlap with the area where subtropical ridges can occur, and the southern part of the Greenland blocking region falls within the North Atlantic area. Please note that the precise definitions of these areas vary slightly in their boundaries between different studies."*

[Figure]

**Figure 1.** Regions over the Euro-Atlantic sector where blocking frequently occurs: Greenland (GL, green box), North Atlantic (N-ATL, blue box), Europe (EUR, orange box), subtropics (SUBTROP, red box) and Ural mountains (URAL, brown box) (Shabbar et al., 2001; Buehler et al., 2011; Luo et al., 2016; Rohrer et al., 2019; Sousa et al., 2021).

L191-203: *"Using the example of an Omega block, Fig. 2 shows possible associated surface extremes depending on the season. Please note that although these extremes are shown schematically in the same plot, they do not necessarily occur simultaneously, even if this is observed in some cases (e.g., Russian heat wave and Pakistan floods in summer 2010). During the cold season (from October to March), low temperature anomalies may be observed in the southern and the eastern parts (at the eastern flank) of the blocking system (Fig. 2 (a)). In addition, there are also cases where snowstorms have been observed at the eastern flank of the blocking system. During the warm season (from April to September), heat waves may develop below the blocking ridge (Fig. 2 (b)). Sometimes these heat waves co-occur with droughts. Moreover, thunderstorms are possible at the eastern and at the western flanks of the blocking system. Heavy rainfall events, which may lead to flooding and which are co-located with areas of high integrated water vapor, are possible at the flanks and near the poleward edge of the blocking ridge during the whole year. The specific location of temperature and precipitation anomalies does however depend on the positioning and type of blocking. For example, Sousa et al. (2021) discuss how different phases of a blocking life-cycle over Western Europe (from an open ridge stage to the posterior stages of Omega and Rex block) during winter impose very distinct regional impacts, a fact the authors explain with the varying morphology of the blocking structure and the corresponding synoptic environment."*

[Figure]

**Figure 2.** Schematic illustration of a blocking system (black line, indicating a geopotential height or PV contour) and some associated surface extremes during a) the cold season (Oct-Mar) and b) the warm season (Apr-Sep). Rossby wave breaking occurs on the flanks of the block, leading to (persistent) cut-off systems in this area. Blue stars show areas where snow storms are observed (eastern flank of the block). Areas with heavy precipitation are marked in light blue (poleward edge of the ridge and at both flanks). Areas with high integrated water vapor transport (IVT) are illustrated in orange. Thunderstorm activity is marked by yellow lightnings. The position of a warm conveyor belt appears in purple. Areas with temperature extremes are marked with dashed lines (red for heat waves, blue for cold spells).

**Minor Comments**

Figure 1: I personally find this figure to be too schematic. What is the authors aim with it? If it is only in the paper to indicate the names of the different areas in use it should be stated so. Otherwise, it might be better to use some figure which gives more realistic representation of blocking regions, potentially also distinguishing between winter and summer (such as figure 1a/e in Davini et al. 2020) In any case, if the authors show the figure is should be discussed and referenced in the text.

Reply: We agree with the reviewer that Figure 1 could be improved and better connected with the text. Please refer to our reply to the main comment above.

Figure 2: Basically the same comment as for figure 1: It is not discussed at all and I am unsure what to take away from it. What are the different impact areas based on? Does a single block have all these effects or are these merely all the potential effects that have been observed/reported at some point? Are they only valid for an omega block in the exact region as indicated or is this to be understood more generally?

Reply: Figure 2 is about "phasing", i.e. where relative to the blocking certain surface extremes occur. This is exemplified by an omega block. So when an omega block is present, these extremes can occur at the marked locations. A good example of this is the case in the summer 2010. There was a heat wave below the blocking ridge (i.e. over Russia/Eastern Europe) and flooding in Pakistan, i.e. associated with the eastern trough (eastern flank of the block). There were also some extreme rainfall events below the western trough (western flank of the block). However, as pointed out by the reviewer, these explanations were missing in the text and have now been added. Please see also our reply to the main comment above.

Some of the shaded areas are quite small (e.g., high IVT and heavy precip to the north), how can they be interpreted? Are there physical mechanisms that can lead to heavy precip only in that area or is it rather that it has just been reported in this area for a specific case?

Reply: We thank the reviewer for the specific questions regarding Figure 2. We have addressed these questions in the revision and have enhanced both the figure and the text. Please see also our reply to the main comment above.

I think it could even be helpful to have several figures with blocks at different locations and their impacts in a more general sense. These could then be referenced in the relevant sections in the text.

Reply: In the previous replies, we have tried to clarify our intentions with this figure. We have chosen the Omega block as the key example for Fig. 2 as it can be used to explain the different influences very well. In general, we agree that dividing the figure into different panels is a good idea. However, we estimate that it is difficult to clearly distinguish the impacts of different blocking types based on the existing literature. This is related to the variety of different indices and definitions. For example, Sousa et al. (2021) discussed the impacts related to different blocking morphologies. We added this reference to the text. Moreover, we have split the figure in terms of seasonal differences, which was suggested by Anonymous Referee #2. Therefore, please also look at the responses to Anonymous Referee #2.

L200-203: *"The specific location of temperature and precipitation anomalies does however depend on the positioning and type of blocking. For example, Sousa et al. (2021) discuss how different phases of a blocking life-cycle over Western Europe (from an open ridge stage to the posterior stages of Omega and Rex block) during winter impose very distinct regional impacts, a fact the authors explain with the varying morphology of the blocking structure and the corresponding synoptic environment."*

section 2.3: Could the authors try to better distinguish the different datasets used to investigate blocking here and in section 2 in general? (or explicitly state whenever statements are valid for simulating blocking in general)

E.g., it is mentioned that blocking representation is a concern in numerical models (line 147) is this referring to global climate models (as discussed in the rest of the paragraph) or also to NWP models? It is further stated that blocking is underestimated but relative to what?

Conversely, are the considerations discussed from line 154 for weather forecast systems also valid for climate models?

Reply: Yes, these statements apply to weather forecast models as well as to climate models, which has now been clarified. The underestimate of blocking compared to observations is clear in both types of model, but the material from line 154 (now line 167) refers to forecast errors and hence to literature specifically dealing with numerical weather prediction.

L160-161: *"The representation of blocking by numerical models has been a longstanding concern in both weather forecasting and climate simulation contexts (e.g., Tibaldi and Molteni, 1990; Pelly and Hoskins, 2003b)."*

216 "separately form each"

Reply: We have revised this sentence.

L243-245: *"However, it is generally difficult to consider the North Atlantic blocking and NAO-separately, as the flow configuration during NAO- itself can be defined as blocking pattern."*

Figure 3: Please make clear that temperature is indicated as shading and gp as lines. Please make clear that dots refer to significance of the temperature anomalies (as I assume).

Reply: We have revised the figure caption as suggested.

P14: *"Figure 3. Monthly 2m temperature (in K, shading) and 500 hPa geopotential height anomalies (in m, contours) based on ERA5 data (Hersbach et al., 2020) (a) in July 2010 (in association with the 2010 heat wave) and (b) in February 2012 (in association with the 2012 cold spell) are shown. Dots mark areas exceeding the 2-sigma level of the 2m temperature climatology (1991-2010)."*

P23: *"Figure 4. Monthly precipitation (in % of the long-term monthly mean, shading) and 500 hPa geopotential height anomalies (in m, contours) based on ERA5 data (Hersbach et al., 2020) (a) in June 2005 (in association with an Iberian drought) and (b) in December 2013 (in association with the 2013 snow storm in the Middle East) are shown. Dots mark areas exceeding (a) the 1-sigma or (b) the 2-sigma level of the precipitation climatology (1991-2010)."*

323: Not sure if the * should be removed from Kautz*?

Reply: Thanks for pointing this out. We have removed the *.

L735-736: *"The forecast variability of the late winter cold spell in March 2018 (see Sect. 3) was investigated by Kautz et al. (2020)."*

421: "surface negative temperature anomalies" should be "negative surface temperature anomalies"?

Reply: We have changed this as suggested.

L468-470: *"For example, cold advection near the eastern flank of a blocking system during winter might lead to widespread negative surface temperature anomalies over the European continent increasing vertical stability (Sillmann et al., 2011; Sousa et al., 2018)."*

435: I acknowledge that the dynamics of precipitation are more complex but I find this paragraph a bit convoluted (in particular compared to, e.g., the one about temperature extremes). Could the authors try to extract clearer high-level impacts here? E.g., it seems a bit strange to me to separately explain the effect of blocking between 0-40E and 0-30E or to switch between clearly defined areas (0-40E) to more general geographical terms ('Central Europe', 'several regions in Europe')

Reply: You are right that this is a very general statement and that local conditions and the exact geographical placement play an important role. This point is discussed in detail in the subsequent paragraphs. To keep the statement more general, we removed the geographical references.

L484-491: *"Links between heavy precipitation and blocking are present across Europe. A high frequency of blocking over Scandinavia results in a high frequency of extreme summer precipitation in Romania and the Eastern Mediterranean (Rimbu et al., 2016) but also in a significant increase of the odds of regional-scale heavy precipitation over Greece and the Central Mediterranean (Lenggenhager and Martius, 2019). Extreme precipitation over Romania is associated with RWB forming downstream of the blocking anticyclones (Rimbu et al., 2016). Blocking over Scandinavia reduces the odds of heavy precipitation over Central Europe, while blocking over the Central and Eastern Atlantic increases the odds of 1-day heavy precipitation in summer over several regions in Europe (Lenggenhager and Martius, 2019). This increase is partly linked to a modulation of cyclone tracks by these blocking systems."*

508: "We next move to blocking related flood cases in Europe" This first example was also about flood in Europe?

Reply: We have chosen this formulation because a thunderstorm case is introduced in the previous paragraph and because we wanted to say that we are now continuing with flooding cases. However, flash floods also occurred in the thunderstorm case, thus, we rephrased this sentence.

L558: *"We now move on to a rain-on-snow flood case study."*

525: "In October 2000 a feedback between heavy precipitation events could be identified." between heavy precip and blocking?

Reply: Yes, we have changed this in the manuscript.

L576-577: *"In October 2000 a feedback between heavy precipitation events and blocking could be identified."*

Figure 4a: The last category is a precipitation anomaly exceeding -100% of the climatology and it seems to exist on the map. This should not be possible, right?

Reply: We agree with the reviewer that there cannot be less than -100% here. The very dark red color is -100%. We have adjusted the color bar so that it no longer looks as if -100% is exceeded somewhere.

[Figure]

**Figure 4.** Monthly precipitation (in % of the long-term monthly mean, shading) and 500 hPa geopotential height anomalies (in m, contours) based on ERA5 data (Hersbach et al., 2020) (a) in June 2005 (in association with an Iberian drought) and (b) in December 2013 (in association with the 2013 snow storm in the Middle East) are shown. Dots mark areas exceeding (a) the 1-σ or (b) the 2-σ level of the precipitation climatology (1991-2010).

572: "low wind conditions" weak wind? or low wind speed conditions?

Reply: We mean "low wind speed conditions", we have changed this in the revised manuscript.

L635-636: *"Both persistent strong and persistent low wind speed conditions can have important impacts, for example, related with the energy production."*

702 "changes blocking occurrence"

Reply: We have corrected this.

L786-787: *"However, changes in blocking occurrence cannot be generalized for the entire Northern Hemisphere, as there are strong regional differences."*

**RC2: Comment on wcd-2021-56 - Anonymous Referee #2**

**https://doi.org/10.5194/wcd-2021-56-RC2, 2021**

The manuscript describes the relationship between atmospheric blocking over the Euro-Atlantic sector and a plethora of extreme events, starting from the more classical heat waves and cold spells up to droughts, extremes of precipitation and compound events.

The discussion is detailed, facing different aspects of both blocking and extreme dynamics, providing a comprehensive state-of-the-art of the scientific knowledge on the topic. Predictability and impacts of climate change are also analyzed.

My main concern is Figures 1 and 2 – the latter is not even referenced in the text! – as they appear as completely disconnected from the main body of the paper. Moreover, they provide much less information that what can be easily achievable with a short climatological/composite analysis.

However, the manuscript provides a useful reference for future studies on the topic, and highlights in which direction the scientific community is showing a lack of knowledge. Therefore, I believe that the manuscript can be easily published in Weather and Climate Dynamics after the suggested revisions are included in the new revision.

Reply: We thank Anonymous Referee #2 for his/her assessment and understand the criticism regarding the embedding of the Figures 1 and 2. As pointed out below, we have revised both figures and referred them in the text.

**Major points**

- As mentioned above, there is no discussion and refencing of Figure 2 in the text: furthermore, Figure 1 is barely described, and the different sectors highlighted in the panel are not analyzed in the text.

Reply: We agree. This point was also criticized by Anonymous Referee #1. We have replaced Figure 1 and have added explanations to the text. Please refer also to our reply to Anonymous Referee #1.

E.g., L228-229: *"Accordingly, blocking, which is characterized by persistent anticyclonic flow anomalies, strongly correlates with the occurrence of European temperature extremes in summer (Fig. 2 (b))."*

- In this direction I believe that Figure 1 will be much more informative if it shows a climatology of atmospheric blocking according to both one reversal and one anomaly index, in a similar fashion to what done by Woollings et al. 2018. This can go hand in hand with a defining, as currently done in Figure 1, a set of "blocking regions", which should be always used in the rest of the manuscript. There is no need of lon-lat definition, but at least something more detailed of "North Atlantic blocking" should be used. Indeed, several times in the text I spotted references to "Atlantic blocking": this is a rather vague entity since it depends on which index is used, and such blurry definition may confound the reader. This is particularly true for this manuscript since we are discussing extremes, where the location of the blocking is fundamental.

Reply: Thanks for this comment. Anonymous Referee #1 has also stated that the informative content of Figure 1 should be increased. We have revised Figure 1 and defined specific blocking regions for which we have provided the corresponding references. For Example, we use these blocking regions in the new overview tables listing selected case studies (Table 1-3).

For example:

**Table 2.** As in Table 1, but for hydrological extremes.

| Type of extreme | Date | Affected region | Blocking region | Damage | References |
|---|---|---|---|---|---|
| drought | 2003, summer | Central, Western Europe | EUR (central) | 70.000 fatalities, losses of €13 billion | Beniston and Diaz (2004), Ogi et al. (2005), García-Herrera et al. (2010), Kron et al. (2019) |
| | 2004/05 | Iberian Peninsula | N-ATL | €1 billion crop damage[1] | Garcia-Herrera et al. (2007) |
| | 2010 | Eastern Europe, Western Russia | SUBTROP, EUR, URAL | 55.000 fatalities, losses of €13 billion | Barriopedro et al. (2011), Lau and Kim (2012) |
| | 2016/17 | Central, Western Europe | SUBTROP | losses of €5.8 billion | Aon (2018) García-Herrera et al. (2019) |
| thunder storm | 2018, May | Central, Eastern Europe | EUR (north) | losses of €380 million | Mohr et al. (2020) |
| flooding | 1954 | Upper Danube | N-ATL (west) | losses of €886 million | Bloschl et al. (2013), Irwin (2016) |
| | 2000, Oct | Southern Alps | N-ATL | 38 fatalities, losses of €7.5 billion | Kron et al. (2019), Lenggenhager et al. (2019) |
| | 2002 | Central Europe | SCAN, EUR (east) | 39 fatalities, losses of €14.5 billion | Bloschl et al. (2013), Kron et al. (2019) |
| | 2011, Oct | Switzerland | N-ATL | losses of €52.5 million | Piaget et al. (2015) |
| | 2013, Jun | Central Europe | SCAN, N-ATL | 25 fatalities, losses of €11 billion | Grams et al. (2014), Kron et al. (2019) |
| snow event | 2013, Dec | Middle East, Germany | EUR (southwest) | 5 fatalities, losses of €106 million (Gaza and West Bank) | Erekat and Nofal (2013), Luo et al. (2015) |
| | 2019, Jan | Alps | N-ATL | 7 fatalities[2] | Yessimbet et al. (2021) |

1 https://www.n-tv.de/panorama/Iberische-Halbinsel-trocknet-aus-article149751.html
2 https://www.bbc.com/news/world-europe-46780856

Similarly, Figure 2 would be much more informative if instead of the current simplified sketch – that is completely disconnected from the current discussion – the authors can provide a composite analysis – based on one or more regions of blocking defined in Figure 1 - bringing together all the dynamical fields they mention. It would be extremely useful if such figure can be divided in both summer and winter, and perhaps if it includes two blocking indices, so that the reader can assess by himself the different nuances of summer and winter blocking and the limitation induced by the blocking index definition (which has been mentioned by the authors in Section 8 has a key issue).

Reply: We agree that Figure 2 was not sufficiently explained in the text. This point was also highlighted by Anonymous Referee #1. Thus, please also see the responses to Anonymous Referee #1, where we also explain our intention with this figure. We would like to keep the figure as simple as possible to provide a good synthesis / overview, but recognize that the distinction between the seasons is a very important point. Therefore, we have split the Figure 2 into two panels so that the extremes for the warm and cold seasons are shown separately.

- Although the manuscript is in general well written, I found some imprecise discussion in the abstract and the introduction. I highlighted some of them in the minor points below, but I recommend the authors to double-check the text and the associated statements.

Reply: We thank Anonymous Referee #2 for highlighting some of these points. We have checked our text again carefully. In some places, we have changed wording and added additional references.

**Minor points**

- L1: "regarding associated impacts". These last words seem not connected to the rest of the sentence, please rephrase.

Reply: We have rephrased this sentence.

L1-2: *"The physical understanding and timely prediction of extreme weather events are of enormous importance to society due to their associated impacts."*

- L9: I might have misunderstood, but why do you mention "longwave radiation warming" under clear sky condition? Perhaps you mean "shortwave" here?

Reply: This was a mistake, thanks for pointing this out. We have changed this to "shortwave radiation".

L8-10: *"In summer, heat waves and droughts form below the blocking anticyclone primarily via large-scale subsidence that leads to cloud-free skies and thus, persistent shortwave radiative warming of the ground."*

- L11: I would say "meridional advection from higher latitudes" or "horizontal advection from continental landmasses". Horizontal advection from high latitude is by definition meridional.

Reply: We have rephrased this.

L11: *"Here, meridional advection of cold air masses from higher latitudes plays a decisive role."*

- L12: The connection between snowfall, blocking and storm track is a bit confusing, I am not sure the three things are robustly related, so that I wonder if it is fundamental to highlight this in the abstract. Extreme snowfall events over Europe are usually associated with easterly or northerly winds of Arctic origin, it is unclear to me what is the role of extratropical cyclones here. Please clarify.

Reply: We have revised these points in the abstract.

L12-13: *"Depending on their location, blocking systems also may lead to a shift in the storm track, which influences the occurrence of wind and precipitation anomalies."*

- L28: The reference to derailed train, although fascinating, does not seem like a relevant information here (no reference is added).

Reply: This sentence was removed.

- L30: please remove "layer up to 10-12 km", the troposphere height is season and latitude dependent.

Reply: We have removed this.

L32-34: *"Namely the prevailing large-scale flow pattern in the troposphere, which was strongly influenced by atmospheric blocking (hereinafter referred to as blocking)."*

- L31: Why blocking is defined as a "self-sustaining tropospheric flow"? Blocking is not a flow – it blocks the flow - but rather an atmospheric pattern or structure.

Reply: We have changed "self-sustaining tropospheric flow features" to "self-sustaining tropospheric flow patterns".

L36-38: *"Blocking systems can be described as long-lasting, quasi-stationary and self-sustaining tropospheric flow patterns that are associated with a large meridional flow component and thus, an interruption and/or deceleration of the zonal westerly flow in the midlatitudes (e.g., Liu, 1994; Nakamura and Huang, 2018)."*

- L37: The plural of blockings is not commonly used in English, while "blocks" is a generally used definition in this case.

Reply: We have corrected this at this point and also at other points in the text.

L40-42: *"In addition, blocking is associated with complex dynamics that link different spatial and temporal scales and affect both their internal evolution and interactions with the flow environment (e.g., Shutts, 1983; Lupo and Smith, 1995)."*

- L88: I would say meridional gradients instead of horizontal gradients, since both the referenced indices uses a meridional gradient.

Reply: We have replaced "horizontal" by "meridional".

L92-94: *"Another way to identify blocking is to detect the reversal in meridional gradients, for example, in the 500 hPa geopotential height (Tibaldi and Molteni, 1990; Scherrer et al., 2006) or the potential temperature at 2 PVU (Pelly and Hoskins, 2003a)."*

- L99: Northern Hemisphere

Reply: We have corrected this at this point and also at other points in the text.

L104-105: *"Blocking in the Northern Hemisphere occurs predominantly for specific regions (Barriopedro et al., 2006; Tyrlis and Hoskins, 2008), both over land and oceans."*

- L165: given that orography has been shown in the last years for being responsible of shaping the mid-latitude flow and having a relevant role in weather and climate model biases, I think this should be mentioned here (e.g., Jung et al 2012, Berckmans et al 2013, Pithan et al. 2016)

Reply: Thanks for the suggestion! We have added this point and these references to this paragraph.

L176-179: *"These include diabatic processes upstream of blocking systems (Rodwell et al., 2013; Quandt et al., 2019; Maddison et al., 2020), orographic effects (Jung et al., 2012; Berckmans et al., 2013; Pithan et al., 2016) and hemispheric Rossby wave teleconnections, often to tropical structures such as the Madden-Julian Oscillation (Hamill and Kiladis, 2014; Parker et al., 2018)."*

- L167: A recent work by Davini et al (2021) on seasonal blocking might be of interest here.

Reply: Thanks for this reference. We have not referenced it in the place mentioned, but rather in Section 6, in a new paragraph focusing on projections with climate models.

L767-768: *"A recent study shows that seasonal forecasts are suitable for analyzing the blocking bias in numerical models, which can help to improve climate models (Davini et al., 2021)."*

- L194: this sentence is a bit strange: a barotropic pressure positive anomaly will lead in the Northern Hemisphere to an anticyclonic circulation: colocation is not a requirement, is a definition. Please rephrase.

Reply: We are not totally sure why this sentence is problematic. We just wanted to say that heat waves occur at the same location as (i.e., are co-located with) barotropic high-pressure anomalies. We slightly rephrased and simplified the sentence to make this clearer.

L226-228: "*Heat waves in Europe are typically co-located with high pressure anomalies and thus anticyclonic circulation conditions throughout the troposphere (Meehl and Tebaldi, 2004; Cassou et al., 2005; Stefanon et al., 2012; Tomczyk and Bednorz, 2016; Zschenderlein et al., 2019).*"

- L246: Why there should be adiabatic compression induced by horizontal advection? Please explain.

Reply: The sentence has been rephrased to make this clearer.

L273-275: "*When the cold air (originating, e.g., in the Arctic region) is transported to the target area, it typically descends, leading to a warming of the air masses due to adiabatic compression and turbulent mixing with warmer air (Bieli et al., 2015).*"

- L248: here – and in other instances, as far as I understand – the authors follow the perspective of an anomaly-based index. This a good choice, but it should be pointed out somewhere in the text that the authors follow this "view" (for this reason I suggest – see main points - showing a blocking climatology in Figure 1 and define a few clear geographical sectors). I would suggest the authors to pay attention to the geographical definition used in the different part of the manuscript, since for example reversal-based blocking indices will show the blocking discussed at these lines over Greenland. Indeed, when using a reversal index blocking in the "North Atlantic" might lead also to a poleward displacement of the jet.

Reply: Since we have not performed new analyses specifically for this paper, we have not limited ourselves to one particular approach. However, we think it would reduce readability if we included in each reference which detection method had been used. Since the dependence of the specific blocking region on the used blocking index is an important point, we have added a sentence about this sensitivity in Section 8.

L861-863: "*The location of the blocking system also depends on the type of detection method (e.g., anomaly vs. reversal of gradient), which is particularly relevant for the phasing between blocking and weather extremes (Doblas-Reyes et al., 2002).*"

- L249: Please remove "in the regions north of these cyclones".

Reply: We have rephrased this sentence.

L277-279: "*Since cyclones moving over Europe can favor the advection of cold continental air masses from northeastern and eastern areas behind their cold fronts, the exact cyclone track has an influence on where a cold spell will potentially develop.*"

- L261-270: this section makes a bit of confusion among seasons. As an example, a warm extreme can be driven by blocking in winter due to advection of warm air from the ocean for a prolonged time. I would encourage the authors to reorganize this part taking the different seasons into consideration.

Reply: Thanks for this suggestion. Our focus is on cold anomalies in winter and warm anomalies in summer. We have revised this in the text.

L289-298: *"As blocking anticyclones are typically embedded in larger-scale Rossby waves, the relationship between temperature extremes and blocking also translates into a linkage of heat and cold spells to Rossby wave activity. European heat extremes often occur in summer during periods of regionally enhanced Rossby wave activity over the Eurasian continent, while winter cold spells in Western Europe and parts of the Mediterranean are more associated with enhanced Rossby wave activity over the North Atlantic (Röthlisberger et al., 2016; Fragkoulidis et al., 2018). The persistence of summer hot spells (winter cold spells) can be increased (decreased) due to recurrent Rossby wave patterns that amplify in the same geographical region (Röthlisberger et al., 2019). Quasi-resonance of hemispheric wave activity (Petoukhov et al., 2013) may lead in summer to simultaneous heat waves in several regions (Kornhuber et al., 2020). Finally, as for other blocking systems (see Sect. 2), the dynamics of anticyclones associated with European summer heat waves can be affected by latent heat release in ascending air masses embedded in upstream wave packets (Zschenderlein et al., 2020)."*

- L305: a brief discussion of marine heat waves and their relationship with blocking might have been interesting here.

Reply: We have added a short paragraph on marine heat waves.

L338-344: *"Marine heat waves can also be related with blocking activity. Regarding the 2003 case, an impact on the sea surface temperatures (SST) of the Mediterranean Sea is documented (Sparnocchia et al., 2006; Olita et al., 2007). The large large-scale blocking over France in 2003 increased air temperature and reduced wind speed (leading to a reduction of all components of the upward heat flux), which were ultimately responsible for the abnormal positive SST anomalies over the Mediterranean Sea. Another example of the importance of blocking activity is the 2012 Northwest Atlantic marine heat wave (Chen et al., 2015b), when persistent atmospheric ridges and blocking through the winter reduced wintertime heat loss from the ocean to the atmosphere (Holbrook et al., 2020)."*

- L324: Why Kautz reference has a *?

Reply: The asterisk was automatically taken from the Bib-file, as the first author L. Kautz and the second author I. Polichtchouk are equally contributing authors in the study. We have remove the asterisk in the text.

L735-736: *"The forecast variability of the late winter cold spell in March 2018 (see Sect. 3) was investigated by Kautz et al. (2020)."*

- L329: I guess that here we are talking about Greenland blocking (Hanna et al. 2016).

Reply: Thanks for this comment. We separate Greenland blocking and North Atlantic blocking as provided in the new Figure 1. However, the southern part of the Greenland blocking region falls within the North Atlantic area. This overlap is addressed in the text.

L118-122: *"The areas shown are partly overlapping. Scandinavian blocking also belongs to the category of European blocking, the southern parts of North Atlantic and European blocking overlap with the area where subtropical ridges can occur, and the southern part of the Greenland blocking region falls within the North Atlantic area."*

L343: what drives the wet anomaly on the eastern flank of the blocking? I can see it coming on the western flank due to the moister low latitude air, but it is a bit unclear to me how this can occur on the downstream side. Is this depending on the geographical placement, i.e., if a blocking is on land or on ocean?

Reply: To answer on a very general level, quasi-geostrophic forcing for lifting and ensuing cyclones and precipitation can be present both upstream and downstream of a blocking anticyclone. We think that the formulation "along the eastern flank" of the blocking may have been confusing as the exact location of the eastern flank depends on whether one considers the anticyclonic flow anomaly only or also the downstream wave-breaking that one typically finds in Rex blocking. We therefore reformulated the sentence.

L388-389: *"Conditions were anomalously dry underneath the blocking anticyclones and anomalously wet to the west and to the east of the blocking anticyclone (cf. Fig. 2)."*

- L480-492: this is another example where a clear geographical region or sector definition may help. It is unclear which kind of blocking episode leads to such dry spell. An "Atlantic blocking" as referred at L487 might have moved the storm track and leads to increase rainfall over Iberia.

Reply: As noted above, the blocking regions have been better defined. We have then used these definitions consistently in the text.

L537-539: *"While the blocking activity within the North Atlantic sector from October to December was similar to the long term average (1958–2005), it was exceptional between January and March, generally the wettest period for the Western Iberian Peninsula."*

- L545: again, it is not very clear here: a high latitude blocking event over the Euro-Atlantic sector might be over Scandinavia so that it can potentially have a limited effect on the storm track.

Reply: Omega blocks favor the poleward shift of the jet "above" the block, while higher latitude Rex blocks tend to divert the jet in two branches, and frequently the southern branch goes to unusual southern latitudes. Santos et al. (2013) provides a comparison between two exceptional winters – 2010 and 2021. They have compared the location of blocking occurrence and the jet position and found that in 2010, there were an equatorward shifted jet and frequent high-latitude blocking while in 2012, a poleward shifted jet and frequent low-latitude blocking were observed. In both cases, the shifted jet has influenced cyclone activity and thus, the occurrence of precipitation anomalies. This study, which we have also cited, underlines that high-latitude blocks have an influence on the storm track.

- L555: does orography – as the Alps - play a role in such configuration?

Reply: For example, the paper of Hofherr and Kunz (2010) provides an extreme wind climatology of winter storms in Germany. They pointed out that over complex terrain, the near-surface wind field is dominated by orography. We have added the reference to the last section.

L834-837: *"This said, it is important to note that not every blocking system leads to the occurrence of an extreme weather event, and that extreme weather events can also be favored by other large-scale flow patterns (such as an intense zonal flow) (e.g., Trigo et al., 2004; Priestley et al., 2017) and strongly influenced by local effects (such as orography) (e.g., Hofherr and Kunz, 2010)."*

- L567: I wonder if this configuration reflects the double wave breaking structure discussed by Messori et al. (2019)

Reply: We thank the reviewer for pointing this out. We have added a sentence and a corresponding reference on this.

L629-630: *"Such large-scale situations are often associated with an intensified jet stream, sometimes sustained by wave breaking on both flanks (e.g. Pinto et al., 2014; Messori and Caballero, 2015)."*

- L624: power plants?

Reply: We have replaced "power plants" by "losses in power plant operation".

L687-688: *"Concurrent droughts and heat waves can cause additional and amplified impacts (e.g., wildfires, crops losses, natural vegetation death, losses in power plant operation, reduction of fisheries) (Zscheischler et al., 2020)."*

- L703: a comprehensive analysis of blocking duration in future scenarios has been done also by Dunn Sigouin et al. (2013)

Reply: Thanks for this reference, which we have added to this paragraph.

L796-798: *"Moreover, the size of blocking systems over the Northern Hemisphere is projected to increase with climate change (Nabizadeh et al., 2019), but no noticeable hemispheric changes are expected in blocking duration (Dunn-Sigouin and Son, 2013)."*

- L706: there are more recent references which analyze and discuss blocking trends, and I think some of the are also referenced in this manuscript (Masato et al 2013, Davini and D'Andrea 2020, etc...)

Reply: Thanks for the additional references. We have added these references to this paragraph.

L:785-786 *"Future projections show generally a decrease in blocking frequency over the mid-latitudes, but there are hints for an increase in blocking duration (Sillmann and Croci-Maspoli, 2009; Davini and D'Andrea, 2020)."*

L789-791: *"In contrast, the frequency of blocking systems over the Euro-Atlantic sector shows a significant decrease in climate simulations in future decades, independent of the considered blocking duration (Matsueda et al., 2009; Masato et al., 2013b)."*

- L730: Screen (2014) might be referenced here.

Reply: Thanks for this reference. We have added this reference here.

L818-819: *"These changes in cold spell characteristics can be partly associated to changes in blocking, whereas changes in other large-scale patterns (zonal temperature gradient and strength of the westerlies) have an additional contribution (Screen, 2014)."*

- Figure3/Figure 4: is this geopotential or geopotential height? Those numbers seem a little too small to me for being m2/s2.

Reply: We thank the reviewer for this question, there is indeed a mistake in the description. We do not show the geopotential in Figure 3 and Figure 4, but the geopotential height. We have corrected the captions accordingly.

P14: *"Figure 3. Monthly 2m temperature (in K, shading) and 500 hPa geopotential height anomalies (in m, contours) based on ERA5 data (Hersbach et al., 2020) (a) in July 2010 (in association with the 2010 heat wave) and (b) in February 2012 (in association with the 2012 cold spell) are shown. Dots mark areas exceeding the 2-sigma level of the 2m temperature climatology (1991-2010)."*

P23: *"Figure 4. Monthly precipitation (in % of the long-term monthly mean, shading) and 500 hPa geopotential height anomalies (in m, contours) based on ERA5 data (Hersbach et al., 2020) (a) in June 2005 (in association with an Iberian drought) and (b) in December 2013 (in association with the 2013 snow storm in the Middle East) are shown. Dots mark areas exceeding (a) the 1-sigma or (b) the 2-sigma level of the precipitation climatology (1991-2010)."*

**Some of the above references:**

Berckmans, J., Woollings, T., Demory, M.-E., Vidale, P.-L. and Roberts, M. (2013), Atmospheric blocking in a high resolution climate model: influences of mean state, orography and eddy forcing. Atmos. Sci. Lett., 14: 34-40. https://doi.org/10.1002/asl2.412

Davini, P, Weisheimer, A, Balmaseda, M, et al. The representation of winter Northern Hemisphere atmospheric blocking in ECMWF seasonal prediction systems. Q J R Meteorol Soc. 2021; 147: 1344–1363. https://doi.org/10.1002/qj.3974

Dunn-Sigouin, E., and Son, S.-W. (2013), Northern Hemisphere blocking frequency and duration in the CMIP5 models, J. Geophys. Res. Atmos., 118, 1179–1188, doi:10.1002/jgrd.50143.

Hanna, E., Cropper, T.E., Hall, R.J. and Cappelen, J. (2016), Greenland Blocking Index 1851–2015: a regional climate change signal. Int. J. Climatol., 36: 4847-4861. https://doi.org/10.1002/joc.4673

Messori, G., et al. On the low-frequency variability of wintertime Euro-Atlantic planetary wave-breaking. Clim Dyn 52, 2431–2450 (2019). https://doi.org/10.1007/s00382-018-4373-2

Pithan, F., Shepherd, T. G., Zappa, G., and Sandu, I. (2016), Climate model biases in jet streams, blocking and storm tracks resulting from missing orographic drag, Geophys. Res. Lett., 43, 7231–7240, doi:10.1002/2016GL069551.

Screen, J. Arctic amplification decreases temperature variance in northern mid- to highlatitudes. Nature Clim Change 4, 577–582 (2014). https://doi.org/10.1038/nclimate2268

**RC3: Comment on wcd-2021-56 – Noboru Nakamura (Referee)**

**https://doi.org/10.5194/wcd-2021-56-RC3, 2021**

**Recommendation: suggested revision**

Comments: This paper surveys current and recent literature describing the weather and climate implications of atmospheric blocking over the Euro-Atlantic sector. Temperature, hydrological, winds and compound anomalies are discussed, together with predictability and future climate projections.

It is widely recognized that blocking patterns drive surface weather anomalies but there are few papers that catalog many specific events that demonstrate their associations and this review paper is unique in that regard. On the other hand, the discussion of hazard types and case studies is sprawled over many disparate events and pertinent publications yet it is hard to grasp key points other than that each event is different (it was a laborious read). Certainly section 8 nicely summarizes the current state of the field and main challenges (I agree with all the points raised in that section), but instead of just laying them out in the conclusion, the authors can proactively structure the text to address some of these challenges.

Reply: We thank Noboru Nakamura for his positive and insightful review. We have now outlined the difficulties and challenges of this area of research in the main text. In section 8, we summarize these challenges, but also address specific points that should be investigated in future research. Following the reviewers' suggestions, we have added short paragraphs addressing specific challenges.

L378-383: *"The development of temperature extremes depends strongly on the persistence and location of blocking. Longer and quasi-stationary blocking periods provide long-lasting favorable conditions for the occurrence of cold spells/heat waves. While the relationship between blocking and temperature extremes is often given, there is a high case-to-case variability both in the phasing and other influences (e.g., soil moisture). Measurement campaigns or sensitivity experiments with numerical models could help to further investigate the multiple interactions. The main challenge here is to cover all relevant components and process chains across a multitude of spatial and time scales"*

L592-597: *"Atmospheric blocking affects the occurrence and persistence of both very dry conditions and extreme precipitation across Europe. However, compared to the link between blocking and temperature extremes, the links between blocking and hydrometeorological extremes are more complex and modulated by local factors such as orography or proximity to moisture sources. Whether such local feedback substantially limit medium-range to sub-seasonal predictability of the blocking related hydrometeorological extremes is an open question. Also a more systematic analysis of the link between blocking and flash floods in summer including the role of soil moisture would be important from a sub-seasonal predictability point of view."*

L695-701: *"In addition to temperature and hydrological extremes, wind extremes can also be influenced by blocking. This influence is primarily caused by changes in the horizontal pressure gradients and/or by a shift in the cyclone tracks. One and the same blocking system can lead to windless conditions in one place and stormy conditions in another. Moreover, blocking can also be responsible for the occurrence of so-called compound events. When multiple extremes are involved, the dynamical interactions within the flow environment of the blocking system become even more complex. This complexity poses a challenge to numerical weather prediction. Our understanding of the complex relationships between blocking, wind extremes and/or compound events would strongly profit for targeted research activities."*

For example:

- I think it is important to stress the importance of case studies at the beginning because (i) the sporadic nature of events hinders statistical analysis of data and (ii) there is a wide variety of weather extremes associated with the types and position of blocks

Reply: Thank you very much for this pertinent suggestion! We agree that the importance of case studies has not yet been sufficiently emphasized in the manuscript. Therefore, we have added a sentence in the introduction.

L60-62: *"In this context, the consideration of case studies is a central issue, as it can best illustrate the complexity and variability in the relationship between blocking and surface extremes."*

- How one generally determines whether a surface event is related to blocking is probably worth discussion before diving into the list of case studies, even though this may reveal the main challenge of the field (metric dependence, etc).

Reply: We agree with this point and that this could be better captured in the text. We have therefore added a paragraph on how studies conclude that a blocking has had an impact on the development/formation of a surface weather extreme event.

L302-305: *"Different approaches were chosen in the studies cited to show this connection: On the one hand, methods such as the calculation of backward trajectories, clustering or correlation analyses were used. On the other hand, there are some studies on surface extreme events in which a synoptic analysis was made, showing that blocking dominated the flow pattern, from which it was assumed that the extreme event was influenced accordingly."*

- It would be useful to have a table (possibly in the supplementary material) that lists the notable events mentioned in this paper, with the dates, affected regions, the types of hazards, the association with the block according to the region specified in Figs.1 and 2, the phase of NAO, and the estimated damage/fatalities. (It is not easy to find an authoritative estimate of economic losses even remotely associated with blocking. The list will be an easy reference for scientists who search for past relevant events.)

Reply: This is an excellent idea. We have added three tables in the revised manuscript: The first table provides an overview of cases of temperature extremes, the second table deals with hydrological extremes. and the third table addresses wind extremes. In these tables, we provide information on damage, blocking region, the region affected by the extreme and corresponding references.

For Example:

**Table 3.** As in Table 1, but for wind extremes.

| Type of extreme | Date | Affected region | Blocking region | Damage | References |
|---|---|---|---|---|---|
| storm | 2007, Jan | Central, Western Europe | EUR (south) | 46 fatalities, insured losses of €4 billion | Fink et al. (2009), Donat et al. (2011) |
| | 2013, Dec | Middle East, Germany | EUR (southwest) | 13 fatalities, losses of €1 billion | Dangendorf et al. (2016), Staneva et al. (2016), Rucińska (2019) |

- If we have a list of events in the table mentioned above, perhaps Sections 3-5 may be shortened, highlighting only quintessential examples.

Reply: We have revised sections 3-5 to increase readability.

E.g., L396-401: *"Droughts have a negative influence on water quantity and quality, thus affecting diverse socio-economic activities and ecosystems. For example, water deficits can lead to crop failure with devastating effects for agriculture (Masih et al., 2014) and influence negatively power generation (Pfister et al., 2006). Dry sommerly conditions may also be favorable for wildfires (Haines, 1989) and other massive air pollution events, with strong impacts on human health (Finlay et al., 2012; Péré et al., 2013; Athanasopoulou et al., 2014). Furthermore, dry spells lead to enhanced soil-atmosphere feedback processes and thus to amplified heat waves (see Sect. 3) (Miralles et al., 2014; Schumacher et al., 2019)."*

**Other points:**

LL17-20 (also Section 3.1): I'm not sure about Europe, but in the US, heat waves on average kill more people annually than any other form of natural hazards: https://www.nrdc.org/sites/default/files/tracking-silent-killer-heat-health-fs.pdf (and many are demonstrably related to blocking). Since heat affects the population in otherwise cool climate the most, its potential danger may be stressed more.

Reply: Thanks for the comment. Also in Europe, heat waves are among the deadliest natural hazards, while storms and floods are among the costliest. We have added further information to the introduction.

L19-20: *"In Europe, heat waves are among the deadliest natural hazards, while storms and flooding events are among the costliest (Kovats and Kristie, 2006; Mohleji and Pielke Jr, 2014; Raška, 2015; Forzieri et al., 2017)."*

LL122-124: This reads like low PV air generated near the surface is advected upward. Is it what it implies? — I suspect latent heating can occur over a column of the troposphere; in that case it is the upward diabatic mass flux that 'dilutes' PV in the upper troposphere that leads to a negative PV anomaly (what gets advected from the boundary is mass, not PV)? (Hayne s and McIntyre 1987, JAS p.828 Fig.2)

Reply: Exactly, we do not want to imply that low PV air is just passively advected upward from the surface. Methven (2015) investigated PV in WCB outflow in general and could show that the PV distribution within a WCB depends primarily on the net diabatic transport of mass. This sentence has been slightly rephrased.

L134-136: "*Latent heating can also contribute to the formation of such negative PV anomalies by enhancing the transport of lower tropospheric air upwards along warm conveyor belts and into the upper anticyclone, where it arrives with low PV values (cf. purple area in Fig. 2) (Madonna et al., 2014; Methven, 2015; Pfahl et al., 2015).*"

LL128-133: Meridional displacement of PV is generally related to Rossby wave transience, but it can operate in different ways — feeding of transient Rossby waves from upstream is an important ingredient but the modulation of quasi-stationary Rossby waves by the remote (sub)tropical sources can be also important.

Reply: Yes, we agree and this information has been added to the text. We have preferred to add this statement in the first paragraph of this section where waves of different timescales are discussed.

L128-129: *"This can include quasi-stationary waves originating in the tropics as well as mid-latitude transients (Hoskins and Sardeshmukh, 1987)."*

LL235-240: Does orography play any role at all (e.g. adiabatic heating associated with a foehn wind)?

Reply: Some studies have provided evidence that warming associated with foehn winds can play a role during heat waves (e.g. Ma et al. 2014). However, since our aim was not to describe the dynamic processes of heat waves in general, but rather to focus on the impact of blocking, we have excluded the information of the orographic effects in this section. However, we have added a more general sentence on orography in the last section.

L834-837: *"This said, it is important to note that not every blocking system leads to the occurrence of an extreme weather event, and that extreme weather events can also be favored by other large-scale flow patterns (such as an intense zonal flow) (e.g., Trigo et al., 2004; Priestley et al., 2017) and strongly influenced by local effects (such as orography) (e.g., Hofherr and Kunz, 2010)."*

LL254-259: Does the balance between radiative cooling and adiabatic warming (subsidence) play more prominent role in summer (in association with heat waves) when advection is weaker?

Reply: Yes, these mechanisms are more important during summer than in winter. In summer, heat waves can develop well underneath (blocking) anticyclones, as the short-wave radiation reaches the ground in an unhindered manner during daytime. In addition, there is adiabatic heating due to subsidence. The radiation-induced cooling then plays a role, especially at night, counteracting the warming through subsidence. As a result, there is a temperature drop during nighttime. However, this temperature drop can be very small - for example during tropical nights when the temperature nevertheless remains above 20 degrees. We have added a short information to the text.

L261-263: *"In addition, this descent is also related to clear-sky conditions that favor surface heating by solar radiation during daytime, counteractive cooling during nighttime, and diabatic heating of the near-surface air through amplified sensible heat fluxes."*

LL349-359: I think droughts here largely refer to meteorological droughts, but there are other types of droughts (hydrological, agricultural, socioeconomic, and ecological) that could spawn from persistent blocking events and making that distinction may be useful.

Reply: Thank you for pointing this out. We have added the used definition of droughts in the revised manuscript.

L401-403: *"In this article, drought refers to meteorological drought, where the atmospheric conditions result in the absence or decrease of precipitation, which in the long run can result in an agricultural drought and/or hydrological drought (Heim Jr, 2002)."*

L482: One of the most exceptional drought —> One of the most exceptional droughts

Reply: We have corrected this.

L533-534: *"One of the most exceptional droughts in the Iberian Peninsula occurred between October 2004 and June 2005 (Fig. 4 (a))."*

LL534-535: Record-breaking snowfall happened the northern part of the Alps —> Record-breaking snowfall happened in the northern part of the Alps

Reply: We have corrected this.

L587: *"Record-breaking snowfall also happened in the northern part of the Alps in January 2019."*

L584: Costal storm surges —> Coastal storm surges

Reply: Corrected.

L647-648: *"Coastal storm surges are typically associated with the passage of a cyclone (or cyclones) near coastal areas, which push the surface water masses towards the coast through wind stress (e.g. Dangendorf et al., 2016)."*

L600: the presence of and a blocking system —> the presence of a blocking system

Reply: We have corrected this.

L661-664: *"Several of the analyzed cases (including the above described storm Xaver, cf. Table 3) point to synoptic situations with a juxtaposition of a passing low pressure center close to Scotland and the presence of a blocking system or an anticyclonic ridge to the south, thus inducting a strong pressure gradient."*

L624: power plants —> power outages (?)

Reply: We have replaced "power plants" by "losses in power plant operation".

L687-688: *"Concurrent droughts and heat waves can cause additional and amplified impacts (e.g., wildfires, crops losses, natural vegetation death, losses in power plant operation, reduction of fisheries) (Zscheischler et al., 2020)."*

LL630-637: The discussion in Section 6 focuses on short-term predictability. But climate models tend to underestimate blocking occurrences in the Euro-Atlantic sector. Does that mean that climate models also underpredict the frequency of extreme weather?

Reply: The representation of blocking systems in climate models was discussed in detail in the review by Woollings et al. (2018). We have added a short paragraph in Section 6. In addition, we consider this topic in section 7, where we point out that it is not possible to transfer the possible trends in blocking frequency to the occurrence of weather extremes. This is partly because weather extremes can also develop without the influence of blocking. Moreover, there are complex interactions on different space and time scales, which make it difficult to make statements about causal relationships under future climate conditions. Thus, one can expect that a change in the frequency of blockings will also have some impact on the occurrence of weather extremes – but there is not enough evidence now to quantify this effect.

L766-774: *"The occurrence and persistence of blocking are underestimated in climate models (cf. Sect. 2) (e.g., Davini and D'Andrea, 2016). A recent study shows that seasonal forecasts are suitable for analyzing the blocking bias in numerical models, which can help to improve climate models (Davini et al., 2021). The underestimation of frequency varies with the region of occurrence and the season (Davini and D'Andrea, 2016). For example, the underestimation of Atlantic/European blocking is lower in summer than in winter (Woollings et al., 2018), which is relevant for the prediction of surface extremes. Studies show that despite the bias, the link between extreme temperatures and blocking can be captured in climate models (Schaller et al., 2018). This is particularly important for heat waves in summer. Nevertheless, as for predictions in the medium and (sub-)seasonal range, the predictability of blocking in climate models cannot be transferred one-to-one to the predictability of surface extremes. In the next section, we discuss the relation between blocking and weather extremes in a warming climate."*